# LGP2 directly interacts with flavivirus NS5 RNA-dependent RNA polymerase and downregulates its pre-elongation activities

Zhongyuan Tan[1☯], Jiqin Wu[2☯], Li Huang[1], Ting Wang[3], Zhenhua Zheng[2], Jianhui Zhang[1], Xianliang Ke[2], Yuan Zhang[2], Yan Liu[2], Hanzhong Wang[2]*, Jianping Tao[1]*, Peng Gong [2]*

1 The Joint Laboratory for Translational Precision Medicine, a. Guangzhou Women and Children's Medical Center, Guangzhou Medical University, Guangzhou, Guangdong, China and b. Wuhan Institute of Virology, Chinese Academy of Sciences, Wuhan, Hubei, China, 2 Key Laboratory of Special Pathogens and Biosafety, Wuhan Institute of Virology, Center for Biosafety Mega-Science, Chinese Academy of Sciences, Wuhan, Hubei, China, 3 The Center for Biomedical Research, Department of Respiratory and Critical Care Medicine, NHC Key Laboratory of Respiratory Disease, Tongji Hospital, Tongji Medical College, Huazhong University of Science and Technology, Wuhan, Hubei, China

☯ These authors contributed equally to this work.
* wanghz@wh.iov.cn (HW); golou6354@163.com (JT); gongpeng@wh.iov.cn (PG)

**Data Availability Statement:** All data generated and analyzed during this study are included in this published article and its supplementary information files.

## Abstract

LGP2 is a RIG-I-like receptor (RLR) known to bind and recognize the intermediate double-stranded RNA (dsRNA) during virus infection and to induce type-I interferon (IFN)-related antiviral innate immune responses. Here, we find that LGP2 inhibits Zika virus (ZIKV) and tick-borne encephalitis virus (TBEV) replication independent of IFN induction. Co-immunoprecipitation (Co-IP) and confocal immunofluorescence data suggest that LGP2 likely colocalizes with the replication complex (RC) of ZIKV by interacting with viral RNA-dependent RNA polymerase (RdRP) NS5. We further verify that the regulatory domain (RD) of LGP2 directly interacts with RdRP of NS5 by biolayer interferometry assay. Data from *in vitro* RdRP assays indicate that LGP2 may inhibit polymerase activities of NS5 at pre-elongation but not elongation stages, while an RNA-binding-defective LGP2 mutant can still inhibit RdRP activities and virus replication. Taken together, our work suggests that LGP2 can inhibit flavivirus replication through direct interaction with NS5 protein and downregulates its polymerase pre-elongation activities, demonstrating a distinct role of LGP2 beyond its function in innate immune responses.

## Author summary

RNA-dependent RNA polymerases (RdRPs) are central components of RNA virus genome replication machinery. Host factors can regulate RNA virus genome replication through direct interactions with RdRPs, typically playing auxiliary roles. LGP2 is a host protein known to play critical roles in innate immune responses and has not been documented in participation of RNA virus genome replication. In this work, we reveal that LGP2 down-regulates flavivirus genome replication through direct interaction with viral

**Funding:** This work was supported by the National Key Research and Development Program (2018YFA0507200 to P.G. and Z.Z.), China Postdoctoral Science Foundation (2020M672579 to Z.T.), National Natural Science Foundation of China (32000136 to J.W.; 32070185 to P.G.), Youth Innovation Promotion Association Program of Chinese Academy of Sciences (2022341 to J.W.), Creative Research Group Program of Natural Science Foundation of Hubei Province, China (2022CFA021 to P.G. and J.W.), and Key Biosafety Science and Technology Program of Hubei Jiangxia Laboratory (JXBS001 to P.G.). The funders had no role in study design, data collection, and interpretation, or submitting the work for publication.

**Competing interests:** The authors have declared that no competing interests exist.

RdRP and its RNA substrate, demonstrating a unique mechanism of RdRP regulation by a host factor.

## Introduction

The *Flavivirus* genus of *Flaviviridae* family includes a large number of arthropod-borne human pathogens, posing a serious threat to human health [1]. The flavivirus genome is a positive-sense RNA that encodes a single polyprotein precursor. The polyprotein is processed into three structural proteins and seven nonstructural proteins (NS1, NS2A, NS2B, NS3, NS4A, NS4B, and NS5) by viral and host proteases [2]. Flavivirus genome replication takes place in an endoplasmic reticulum (ER) membrane-associated viral replication complex (RC) including viral non-structural proteins, viral RNA, and host proteins [3]. NS5 and NS3 are central components of RC on a scaffold created by the other five transmembrane proteins, and they are together responsible for all enzymatic activities required to amplify the viral RNA [4]. In the first step of genome replication, the positive-sense genomic RNA serves as the template, and the negative-sense RNA is synthesized by NS5 RNA-dependent RNA polymerase (RdRP) module, forming a double-stranded RNA (dsRNA) replication intermediate. The negative-sense RNA then serves as the template for NS5 to synthesize adequate positive-sense RNA, which is 5′-capped and methylated with the cooperation of NS5 methyltransferase (MTase) and NS3 NTPase/triphosphatase modules [5]. NS3 also possesses RNA helicase and protease functions, playing key roles in resolving RNA tertiary structures during RdRP synthesis and viral polyprotein processing [6]. Therefore, NS5 and NS3 play critical roles in flavivirus replication and their dysfunction may strongly impact the virus life cycle.

It has been documented that some host proteins involved in the viral RC may provide auxiliary functions related to efficiency and specificity for flavivirus replication. For example, reticulon 3.1 (RTN 3.1A) [7], fatty acid synthase (FASN) [8], heat shock protein 40 (HSP40) chaperon protein DNAJC14 [9], vimentin [10], transmembrane protein 41B (TMEM41B) [11], and atlastins (ATLs) [12] are recruited to RC through interactions with viral proteins. These host proteins mainly facilitate RC membrane shaping and offer a scaffold for RC central components NS5 and NS3. However, a few host proteins also directly interact with NS5. Tripartite motif (TRIM) protein TRIM79α [13], signal transducer and activator of transcription 2 (STAT2) [14], Golgi brefeldin A resistance factor (GBF1) [15], cyclophilin A (CyPA) [16], and protein kinase G (PKG) [17] interact with flavivirus NS5 and regulate viral replication. However, host proteins that are directly involved in RC and downregulate viral genome replication have not been well documented in flaviviruses or even in all positive-strand RNA viruses.

Retinoic acid-inducible gene-I (RIG-I)-like receptors (RLRs), including RIG-I, melanoma differentiation antigen 5 (MDA5), and laboratory genetics and physiology 2 (LGP2), are cytoplasmic viral RNA sensors that recognize and bind viral dsRNA to trigger antiviral innate immune responses [18]. They all comprise of a DExD/H helicase domain and a C-terminal domain (CTD) that is also known as regulatory domain (RD) [19]. The helicase module has ATPase activity and the RD domain plays key roles in RNA recognition and binding, as supported by studies characterizing purified RLRs RD domains *in vitro* [20], and LGP2 binds to dsRNA or single-stranded RNA (ssRNA) with higher affinity than either RIG-I or MDA5 [21,22]. However, lacking the N-terminal two caspase activation and recruitment domains (CARDs) compared with RIG-I and MDA5, LGP2 only functions as a regulator of RIG-I and MDA5, not directly triggering RLR signaling pathway by activating the downstream adaptor protein mitochondrial antiviral signaling (MAVS) [19]. Previous studies suggest that LGP2

negatively or positively regulates RIG-I and MDA5 in response to infection of different viruses [23,24]. For example, LGP2 inhibits IFN-stimulated regulatory element- and NF-ĸB-dependent pathways induced by Sendai virus (SEV) and Newcastle disease virus (NDV) infection [25]. Hepatitis C virus (HCV) infection promotes the interaction of LGP2 with MDA5 and facilitates MDA5 recognition of HCV RNA that induced IFN signaling [26]. The transgenic mouse overexpressing LGP2 decreases inflammatory mediators and leukocyte infiltration into the bronchoalveolar airspace during influenza A virus (IAV) infection [27]. The production of IFNβ significantly decreases in response to various RNA viruses, such as encephalomyocarditis virus (EMCV), mengovirus, Japanese encephalitis virus (JEV), SEV, and vesicular stomatitis virus (VSV), in an LGP2-knockout cell line [28]. Therefore, LGP2 plays an important role in RLRs signaling in response to various viral infections. Except for acting as a regulator of RIG-I and MDA5, LGP2 is also involved in other processes by interacting with key proteins of virus or host. For example, LGP2 binds to the dsRNA binding sites of TAR-RNA binding protein (TRBP), resulting in inhibition of pre-miRNA binding and recruitment by TRBP [29]. The leader protease (Lpro) of foot-and-mouth disease virus (FMDV) interacts with LGP2 and targets LGP2 for cleavage at the an RGRAR sequence [30].

Zika virus (ZIKV)/dengue virus (DENV) and tick-borne encephalitis virus (TBEV) are representatives of mosquito- and tick-borne flaviviruses, respectively [31,32]. Both ZIKV and TBEV can infect central nervous system (CNS) and replicate in multiple types of neural cells and abrogate neurogenesis, with ZIKV causing microcephaly in newborns and Guillain-Barre syndrome [33] and TBEV causing encephalitis [34]. DENV infection can lead to fatal dengue hemorrhagic fever/dengue shock syndrome (DHF/DSS) [35]. Using ZIKV, DENV, and TBEV systems in this study, we find that LGP2 colocalizes with viral RC and directly interacts with the key replication protein NS5, and inhibits its polymerase function. This function is distinct from previously documented RLR-related functions in innate immune response, suggesting a unique route of antiviral regulation of LGP2 by directly targeting viral genome replication machinery.

## Results

### LGP2 affects ZIKV replication in astrocytes when knocked down or overexpressed

A previous study indicates that ZIKV infection generates RIG-I- and MDA5-stimulatory RNAs in human non-small cell lung cancer cell line (A549) [36]. Here we found that ZIKV infection also upregulated RIG-I and MDA5 at mRNA and protein levels (S1A, S1C and S1D Fig) when human astrocytoma cells (CCF-STTG1) were infected by the virus. It is not surprising that these two RLRs are upregulated in CCF-STTG1 cells, because astrocytes are important immune response cells in various neurotropic viral infections to respond to CNS injury and degenerative diseases [37]. It was reported that ZIKV induces a robust antiviral response in a manner dependent on RIG-I and MDA5 in human astrocytoma cells [38]. Interestingly, we found that LGP2 is also upregulated at mRNA and protein levels (S1A and S1B Fig) in CCF-STTG1 cells. Therefore, we hypothesized that LGP2 could play a key regulatory role during ZIKV infection in astrocytes. SiRNA-mediated knockdown of LGP2 in CCF-STTG1 cells was detected at mRNA and protein levels (Fig 1A and 1B). Then, we infected CCF-STTG1 cells with ZIKV, and the results indicate that the knockdown of LGP2 enhances ZIKV RNA replication at 24 and 36 hours post-infection (hpi) (Fig 1C), and enhances ZIKV titer at 24 and 36 hpi (Fig 1D). We further hypothesized that LGP2 could inhibit ZIKV proliferation in CCF-STTG1 cells. To dissect the function of LGP2, we performed ZIKV infection in LGP2-o-verexpressing CCF-STTG1 cell line. The results indicate that lentiviral vector-based LGP2

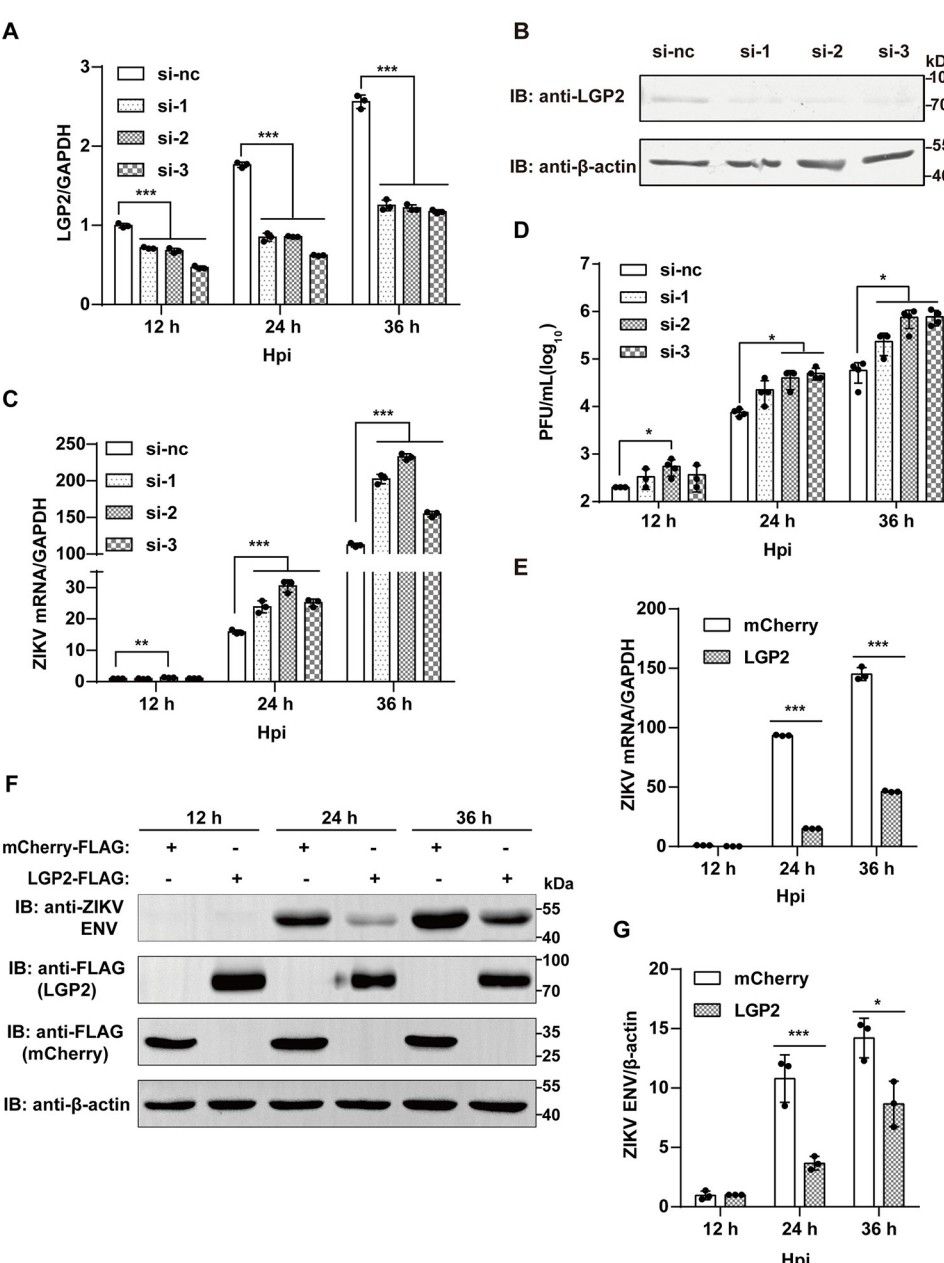

**Fig 1. SiRNA-mediated knockdown and overexpression of LGP2 affect the replication of ZIKV.** (A-D) The LGP2-knockdown CCF-STTG1 cells were infected by ZIKV (MOI = 0.1), and RT-qPCR and virus titer samples were obtained at 12, 24, and 36 hpi. Western blot samples were obtained at 24 hpi. (A) The relative mRNA levels of LGP2 quantified by RT-qPCR. (B) The protein expressions of LGP2 and β-actin determined by Western blot. (C) The relative quantification of ZIKV genomic RNA. (D) Virus titers of the cell supernatants measured by virus plaque on Vero cells. (E-G) LGP2- and mCherry-overexpressing CCF-STTG1 cells were infected by ZIKV (MOI = 0.1). RT-qPCR and Western blot samples were obtained at 12, 24, and 36 hpi. (E) The relative quantification of ZIKV genomic RNA. (F) The protein expressions of LGP2, mCherry, ZIKV ENV, and β-actin determined by Western blot. (G) The bands were analyzed by ImageJ and fold changes of ZIKV ENV/β-actin were calculated and shown. Data collected from three independent experiments were shown as Means ± SD (Student's t-test; *: p<0.05, **: p<0.01, ***: p<0.001).

overexpression inhibits the ZIKV replication at 24 and 36 hpi by RT-qPCR (Fig 1E) and Western blot (Fig 1F and 1G) analyses. In summary, we find that ZIKV infection upregulates LGP2 production and LGP2 in turn inhibits ZIKV replication in astrocytes.

## LGP2 inhibits ZIKV and TBEV replication independent of IFN induction

LGP2 acts as a negative regulator for IFN signaling and proinflammatory cytokines by interfering with RIG-I and tumor necrosis factor (TNF) receptor-associated factor (TRAF) family proteins [20,29]. To explore the function of LGP2 in regulating innate immune response during ZIKV infection, the mRNA level of IFNβ, interferon stimulated genes (ISGs) ISG56/myxovirus resistance protein 1 (Mx1), NFκB inhibitor α (IκBα), and proinflammatory cytokines interleukin 6 (IL6)/C-X-C motif chemokine 10 (CXCL10) was quantified by RT-qPCR in the LGP2-knockdown CCF-STTG1 cells during ZIKV infection (S2 Fig) [29,38]. The results indicate that LGP2 knockdown (siRNA si-2 shows the best inhibition effect was chosen) significantly increases the mRNA levels of IFNβ, ISG56, Mx1 and IL6 at multiple post infection time points (S2A and S2C–S2E Fig), and IκBα and CXCL10 at 24 hpi (S2B and S2F Fig). In parallel, we measured the mRNA levels of these genes in the LGP2-overexpressing CCF-STTG1 cells during ZIKV infection. The results indicate that LGP2 overexpression significantly decreases the mRNA levels of all these genes at 24 and 36 hpi (S3 Fig). Taken together, LGP2 suppresses the IFN signaling and proinflammatory cytokines during ZIKV infection that is consistent with previous studies [20, 29, 39]. We hypothesized that LGP2 could inhibit ZIKV replication independent of IFN signaling and proinflammatory cytokines.

Capable of RNA replication but not virus particle assembly, a replicon is an ideal tool for characterization of the replication of viral genome RNA. We therefore constructed the ZIKV and TBEV replicons that contained the *Renilla* luciferase (Rluc) reporter gene (Fig 2A), and then transfected LGP2-overexpressing plasmids and replicons in human embryonic kidney cell line (293T). Firstly, we verified LGP2's inhibitory effect on IFN induction using well-established model virus SEV (S4 Fig). Under this condition, luciferase assay data indicate that the replication of ZIKV (Fig 2B) and TBEV (Fig 2C) replicons is inhibited by LGP2 overexpression. These results demonstrate that LGP2 inhibits the genome replication of both mosquito-borne ZIKV and tick-borne TBEV, suggesting the related mechanism may be conserved in flaviviruses.

A previous study shows that two LGP2 mutants with deletions to motifs IV and V (MIV and MV) no longer suppress IFN signaling response to SEV infection in HeLa and 2fTGH cells [40]. However, we found that these two LGP2 mutants were still capable of inhibiting IFN induction in SEV-infected 293T cells, albeit with lower inhibition levels (S4 Fig). We overexpressed the two LGP2 mutants (MIV and MV) and ZIKV replicon in 293T cells. Luciferase assay data indicate that LGP2 MIV and MV mutants also inhibit the replication of ZIKV replicon, consistent with the LGP2 wild-type (WT) behavior (Fig 2D and 2E). We next tested LGP2 functions in Vero cell line that are kidney epithelial cells isolated from a normal African green monkey and are unable to synthesize IFN and mount an IFN-dependent antiviral response [41]. Luciferase assay data indicate that LGP2 also inhibits the replication of ZIKV (Fig 2F) and TBEV (Fig 2G) replicons by overexpressing LGP2 in Vero cells. These observations indicate that LGP2 still inhibits ZIKV and TBEV replication in an IFN-deficient system. Therefore, we hypothesized that LGP2 could inhibit ZIKV and TBEV replication through a distinct way beyond conventional IFN induction.

## LGP2 colocalizes with the replication complex of ZIKV

Bir A is a 35-kD DNA-binding biotin protein ligase in *Escherichia coli* (*E. coli*) that regulates the biotinylation and acts as a transcriptional repressor for biotin biosynthetic operon. Bir A

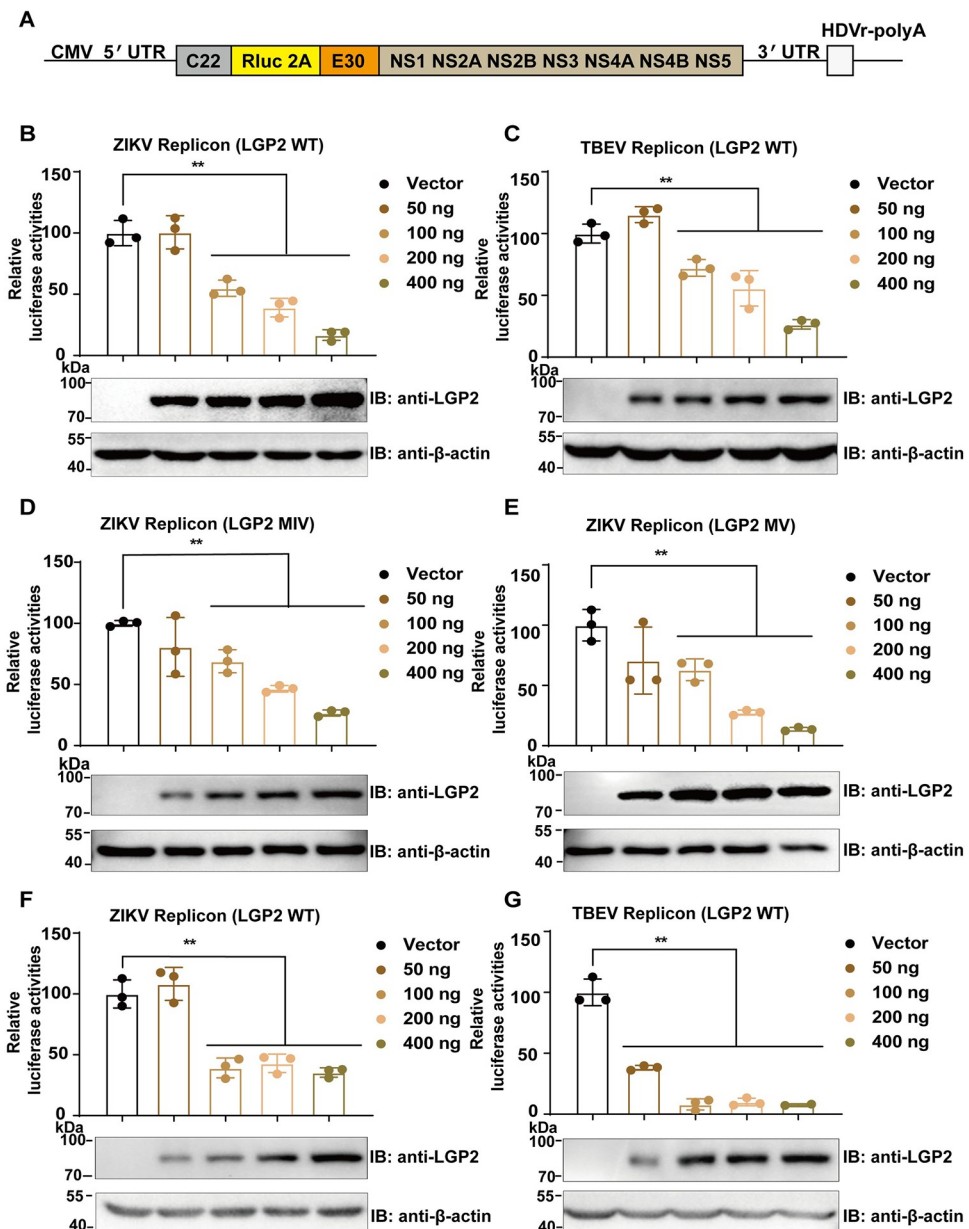

**Fig 2. LGP2 inhibits ZIKV and TBEV replication independent of IFN induction.** (A) Schematic diagram of TBEV and ZIKV replicons. (B-E) 0, 50, 100, 200, 400 ng LGP2 WT- or its mutant (MIV and MV)-overexpressing plasmids were co-transfected with 100 ng viral replicons in 293T cells. Vector plasmid (pCAGGS-HA) was added to ensure the equally total amount (500 ng) of transfected plasmid DNA in each sample. Luciferase activities were detected and relative luciferase activities of replicon transfected with vector were set to 100%. Relative luciferase activities of ZIKV replicon (B) and TBEV replicon (C) upon overexpression of LGP2 WT. Relative luciferase activities of ZIKV replicon upon overexpression of LGP2 mutants (MIV (D) and MV (E)). (F-G) LGP2-overexpressing plasmids were co-transfected with viral replicons in Vero cells. Relative luciferase activities of ZIKV replicon (F) and TBEV replicon (G) upon overexpression of LGP2 WT in Vero cells. Relative luciferase activities of replicon transfected with vector were set to 100%. The protein expressions of LGP2 were determined using Western blot. Data collected from three independent experiments were shown as Means ± SD (Student's t-test; **: p<0.01).

mutant (R118G, Bir A*) can result in promiscuous protein biotinylation because its free bioAMP readily reacts with primary amines [42]. In order to identify LGP2 interacting viral proteins, we constructed the transfer plasmid of fusion protein LGP2-BirA*, and packaged

lentiviruses in 293T cell line by using a third-generation lentivirus packaging system. The stably LGP2-BirA*-overexpressing cell line was established using the lentiviruses to infect CCF-STTG1 cells. Then, we infected the stably overexpressing cell line with ZIKV and cultured it in 1640 culture medium with 50 μM biotin. Cell lysates were incubated with streptavidin-conjugated magnetic beads, immunoprecipitated by the biotinylated protein, and detected by Western blot using horseradish peroxidase (HRP)-streptavidin or ZIKV NS5/NS3 antibodies. Western blot data indicate that ZIKV NS5 and NS3 were biotinylated by LGP2-BirA* and immunoprecipitated by magnetic beads. These results demonstrate that LGP2 is spatially adjacent to ZIKV NS5 and NS3 (Fig 3A). Our data therefore suggest that LGP2 may be an important host protein regulating ZIKV RC.

Flavivirus NS3-NS5 interactions have been documented in several studies [43,44]. In order to understand the relationship between NS3, NS5, and LGP2, we performed a competitive Co-IP assay. The results indicate that the change of ZIKV NS3 amount does not much affect the Co-IP of LGP2 and ZIKV NS5, but change of LGP2 amount has an apparent effect on the Co-IP of ZIKV NS3 and NS5 (Fig 3B). These data suggest that LGP2 can impair NS3 binding to NS5, but not vice versa. Confocal immunofluorescence microscopy was further used to analyze the cellular localization of LGP2, NS5, and dsRNA. HeLa cells were transfected with LGP2-GFP-overexpressing plasmids and infected with ZIKV, and then the cellular localization of LGP2, NS5, and dsRNA was examined. LGP2-GFP and dsRNA were localized exclusively in the cytoplasm, whereas NS5 was localized both in the cytoplasm and nuclear regions. In the two-channel colocalization images, the colocalization of LGP2-dsRNA, ZIKV NS5-dsRNA and LGP2-ZIKV NS5 are visualized in the cytoplasm (Fig 3C) with Manders' colocalization coefficients (MCCs) greater than 0.5 (Fig 3D). In the three-channel colocalization image, the colocalization of LGP2-dsRNA-ZIKV NS5 is also visualized in the cytoplasm around cell nucleus (Fig 3C) with MCCs of each two channels greater than 0.5 (Fig 3E). These data further indicate that LGP2 colocalizes with ZIKV NS5 and dsRNA that are both key components of the ZIKV RC. Altogether, these data demonstrate the colocalization of LGP2 and RC during ZIKV infection, and LGP2 could possibly interact with NS5.

## LGP2 may interact with the RdRP module of NS5

To further dissect flavivirus RC regulation by LGP2, LGP2 was overexpressed in A549 cells, and then the cells were infected by ZIKV. An endogenous Co-IP assay showed that LGP2 could Co-IP with endogenous ZIKV NS5 (Fig 4A). Therefore, these results show that the interaction of LGP2 with ZIKV NS5 could occur in ZIKV infection. Meanwhile, a series of Co-IP assays were performed by co-transfection. These data indicate that LGP2 also co-immunoprecipitates with TBEV NS5 (Fig 4B). We next compared the Co-IP of LGP2 with NS5 RdRP or MTase modules, respectively. The results indicate that LGP2 only co-immunoprecipitates with the RdRP but not the MTase (Fig 4C). Since both LGP2 and NS5 can bind to RNA, the possible interaction between LGP2 and NS5 may be mediated by RNA bridging. We therefore compared the Co-IP results with and without the treatment of the cell lysates by a mixture of two single-stranded RNA ribonucleases RNase A and RNase I, and the results indicate that RNA degradation by this RNase mixture does not affect the Co-IP of LGP2 and NS5, but weakens the Co-IP of LGP2 and Dicer (Fig 4D) [45]. These Co-IP data together suggest that LGP2 is involved in viral RC and likely interact with NS5 RdRP module and viral RNA.

## The regulatory domain of LGP2 interacts with NS5

To verify the critical NS5 binding site of LGP2, we performed a series of truncations at LGP2 C-terminus (537–678 residue) (Fig 5A), and used Co-IP and luciferase assay experiments to

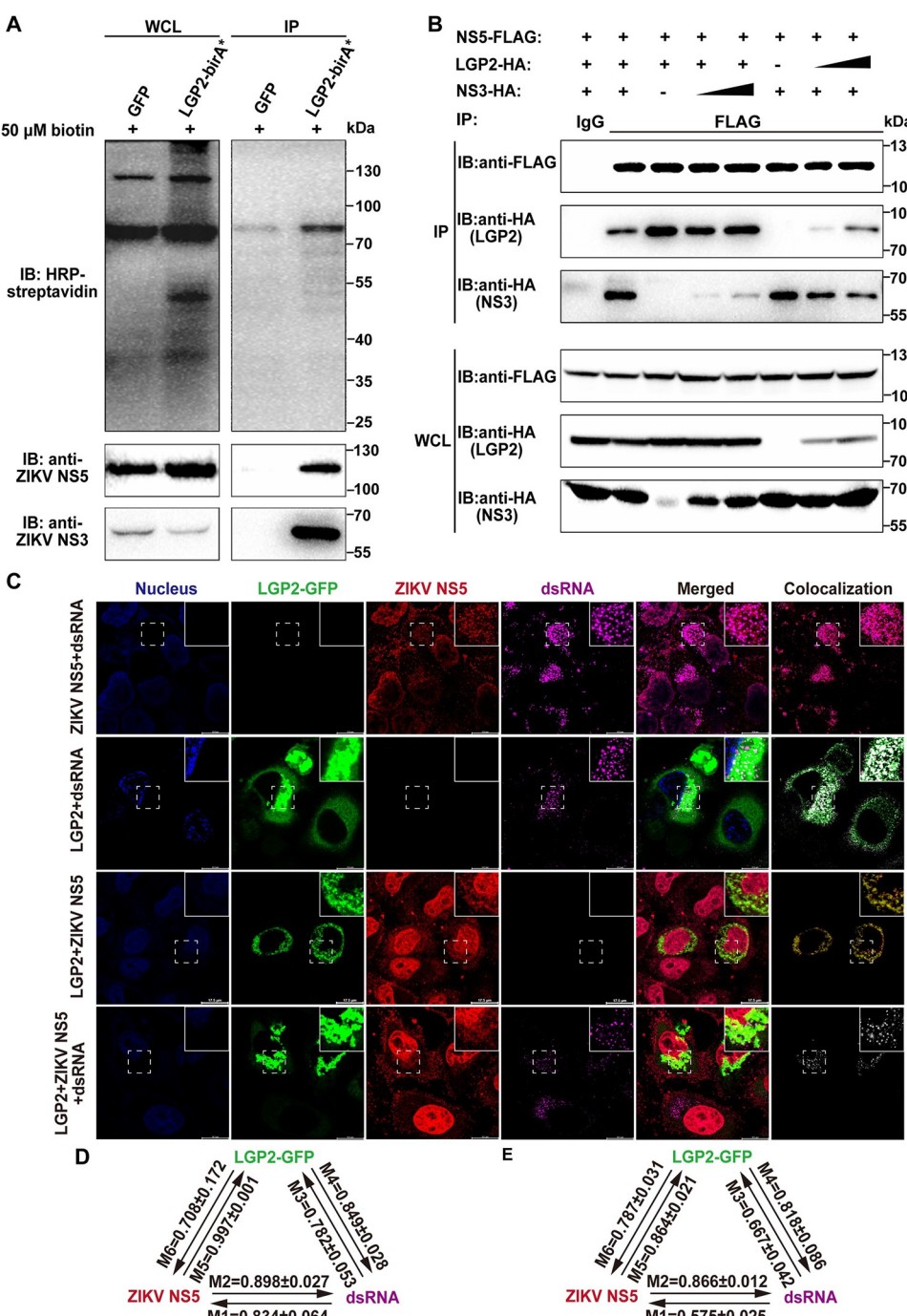

**Fig 3. LGP2 colocalizes with the RC of ZIKV.** (A) The affinity capture of biotinylated viral proteins. LGP2-BirA*-overexpressing CCF-STTG1 cells were infected by ZIKV (MOI = 0.1) and incubated in complete media supplemented with 1 μg/mL puromycin and 50 μM biotin for 48 h. The supernatants of cell lysates were used for Co-IP and Western blot analysis. Three independent experiments were performed and images from one experiment were shown. (B) A competitive Co-IP assay of LGP2 to NS5 and NS3. 2 μg LGP2 and (0, 0.5, 1 μg) ZIKV NS3, or 2 μg ZIKV NS3 and (0, 0.5, 1 μg) LGP2-overexpressing plasmids were transfected into 293T cells with 2 μg ZIKV NS5-overexpressing plasmids. The supernatants of cell lysates were used to perform Co-IP analysis, and immunoprecipitates were analyzed by Western blot. Three independent experiments were performed and images from one experiment were shown. (C) Cellular localizations of LGP2, ZIKV NS5, and viral dsRNA during ZIKV infection. HeLa cells were transfected with LGP2-GFP-overexpressing plasmids and infected by ZIKV (MOI = 5) after 16–20 h. At 24 hpi, the cells were fixed and analyzed by confocal immunofluorescence. Cellular nuclei (Blue), LGP2-GFP fusion protein (Green), ZIKV NS5 (Red) and viral dsRNA (Purple) were stained and excited with 405/488/555/633 nm lasers, respectively. Image processing

and colocalization analysis was performed by LAS X and Imaris. The representative four channels, merged, and colocalization images were shown, and the insets at the top-right corner of each images showed the magnified 2× view of the area marked and the scale bars showed at the bottom-right corner. (D) Pairwise MCCs of two-channel in (C) (ZIKV NS5+dsRNA, LGP2+dsRNA, LGP2+ZIKV NS5) were shown. (E) Pairwise MCCs of two-channel each other in (C) (LGP2+ZIKV NS5+dsRNA) were shown. M1 = ZIKV NS5+dsRNA/ZIKV NS5, M2 = ZIKV NS5+dsRNA/dsRNA, M3 = LGP2+dsRNA/LGP2, M4 = LGP2+dsRNA/dsRNA, M5 = LGP2+ZIKV NS5/ LGP2, M6 = LGP2+ZIKV NS5/ ZIKV NS5. At least 10 cells were randomly selected for analysis.

evaluate the impact of these truncations on NS5 interaction and virus replication. We found that the LGP2 1–646 mutant can still interact with NS5 but its ability to inhibit viral replication is weakened. LGP2 1–596 mutant can no longer interact with NS5 and its ability to inhibit viral replication is decreased if compared with LGP2 WT (Fig 5B and 5C). These results suggest that the primary interaction sites of LGP2 with NS5 is likely within the 596–678 region of RD and the interaction plays a key role in the inhibition of viral replication.

To further verify the interaction of LGP2 with NS5, we purified the RD of LGP2 (LGP2 RD, residues 537–678) [21] as well as ZIKV/TBEV/dengue virus serotype 2 (DENV2) NS5 proteins, and used the biolayer interferometry to measure the equilibrium dissociation constant ($K_d$) of LGP2 RD and NS5. The association and dissociation processes between immobilized LGP2 RD and flowing ZIKV NS5 (Fig 5D), TBEV NS5 (Fig 5E), DENV2 NS5 (Fig 5F) or human serum albumin (HSA, negative control) (Fig 5G) at a series of concentrations were monitored in real time. ZIKV, TBEV, and DENV2 NS5 produced enough optical response to indicate the interaction with LGP2 RD, while HSA did not. The $K_d$ values were 88±11 nM, 1.2±0.8 μM, and 75±14 nM for RD-ZIKV NS5, RD-TBEV NS5, and RD-DENV2 NS5 interactions, respectively. These results again support direct interaction between LGP2 RD and flavivirus NS5.

## LGP2 inhibits RdRP activity of NS5

To understand the impact of LGP2 RD on the polymerase function of NS5, we carried out an *in vitro* RdRP assay to dissect the underlying mechanisms following methods established in our DENV2 and TBEV NS5 studies, and we chose DENV2 instead of ZIKV because we have established RdRP assays in the former [46,47]. LGP2 RD was provided at different molar ratios to NS5 and a pre-incubation was performed to ensure its binding to NS5. Then, ATP and UTP were provided as the only NTP substrates for the production of 9-mer product (P9) directed by a T30/P2 RNA construct comprising a 30-mer template (T30) and a G-G dinucleotide primer with a 5′-monophosphate (P2) (Fig 6A). We performed the *in vitro* RdRP assay as shown in Fig 6B. The results indicate that the amount of P9 products synthesized by NS5 is decreased when LGP2 RD is added and is positively correlated with the amount of LGP2 RD added for both DENV2 (Fig 6C, compare lanes 7–8, 13–14 to 10–11, 16–17, respectively) and TBEV NS5 (Fig 6D, compare lanes 1, 3, 7–9, 13–15 to 4, 6, 10–12, 16–18, respectively). Intercellular adhesion molecule 1 (ICAM-1) acting as the negative control protein can't inhibit P9 products synthesized by DENV2 NS5 (S5A Fig). These data suggest that LGP2 RD can inhibit polymerase activities of DENV2 and TBEV NS5. Since LGP2 is likely not very abundant in viral RC, the relatively high molar ratio (5:1) of LGP2 RD:NS5 required for observation of the inhibitory effect in DENV2 highlights the difference between the RdRP assay and *in vivo* situation, with respect to components involved in the RNA synthesis and solution conditions.

## LGP2 inhibits flavivirus RdRP in pre-elongation stages

To further dissect the mechanism of LGP2 RD inhibition of NS5 RdRP synthesis, we investigated LGP2 RD effect at polymerase pre-elongation and elongation stages in two different assays. In the DENV2 system, an elongation complex (EC) is formed upon the synthesis of the

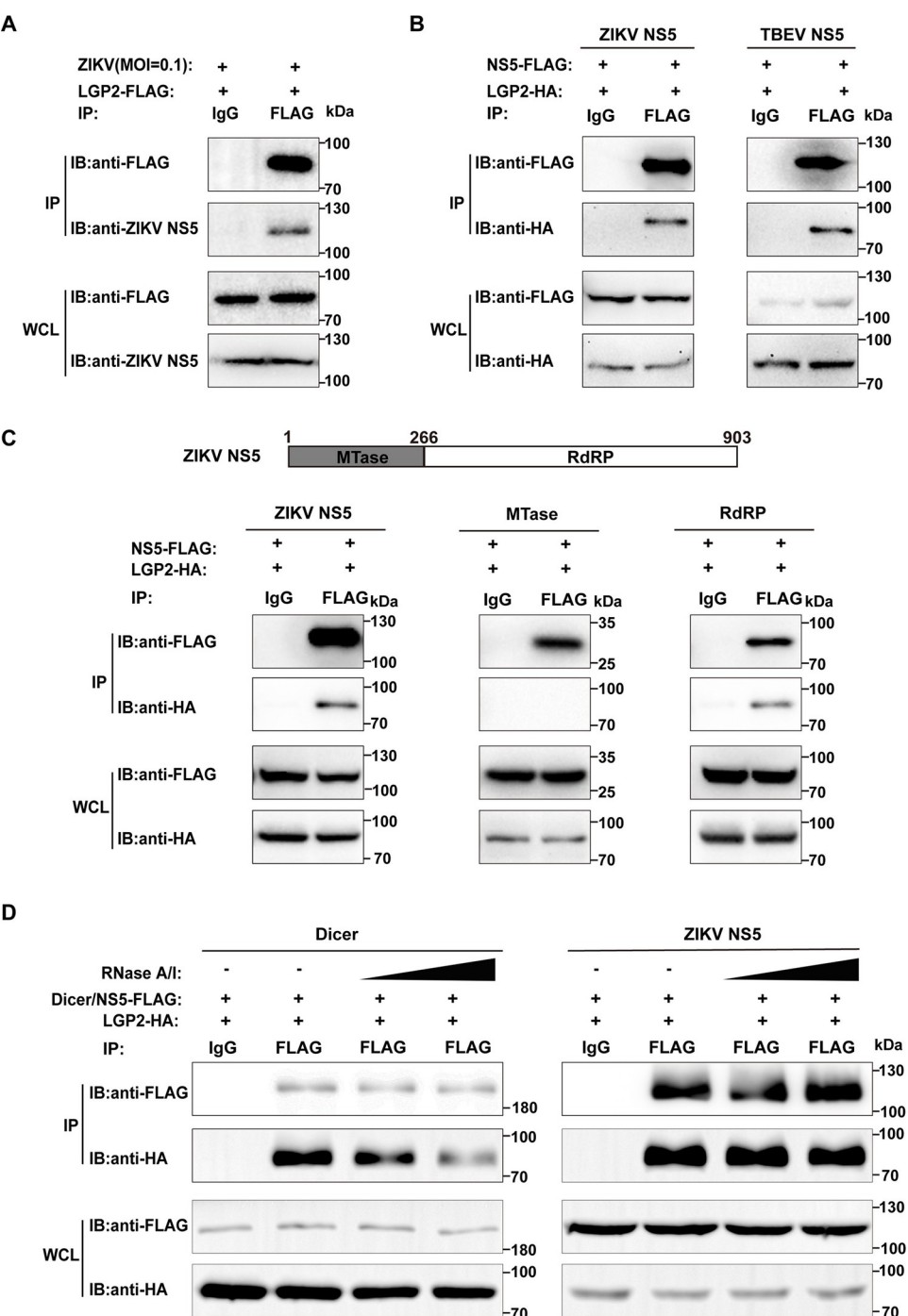

**Fig 4. LGP2 interacts with the RdRP module of NS5.** (A) A Co-IP assay of LGP2 and ZIKV NS5 during viral infection. LGP2-overexpressing plasmid was transfected into A549 cells and infected with ZIKV (MOI = 0.1) after 16–20 h. At 48 hpi, the supernatants of cell lystates were used for Co-IP analysis. (B-C) A series of Co-IP assays of ZIKV NS5, TBEV NS5, MTase module, or RdRP module with LGP2 by plasmid co-transfection in 293T cells. After 24 hpi, the supernatant of the cell lysates was used for Co-IP analysis. (D) A Co-IP assay of LGP2 and ZIKV NS5 with treatment of an RNase A-RNase I mixture. Two plasmids co-transfected into 293T cells, and the supernatants of cell lysates were pre-treated by the RNase mixture prior to Co-IP analysis. All of the Co-IP samples were analyzed by using Western blot. A Co-IP assay of LGP2 and Dicer was performed as a positive control. Three independent experiments were performed and images from one experiment were shown.

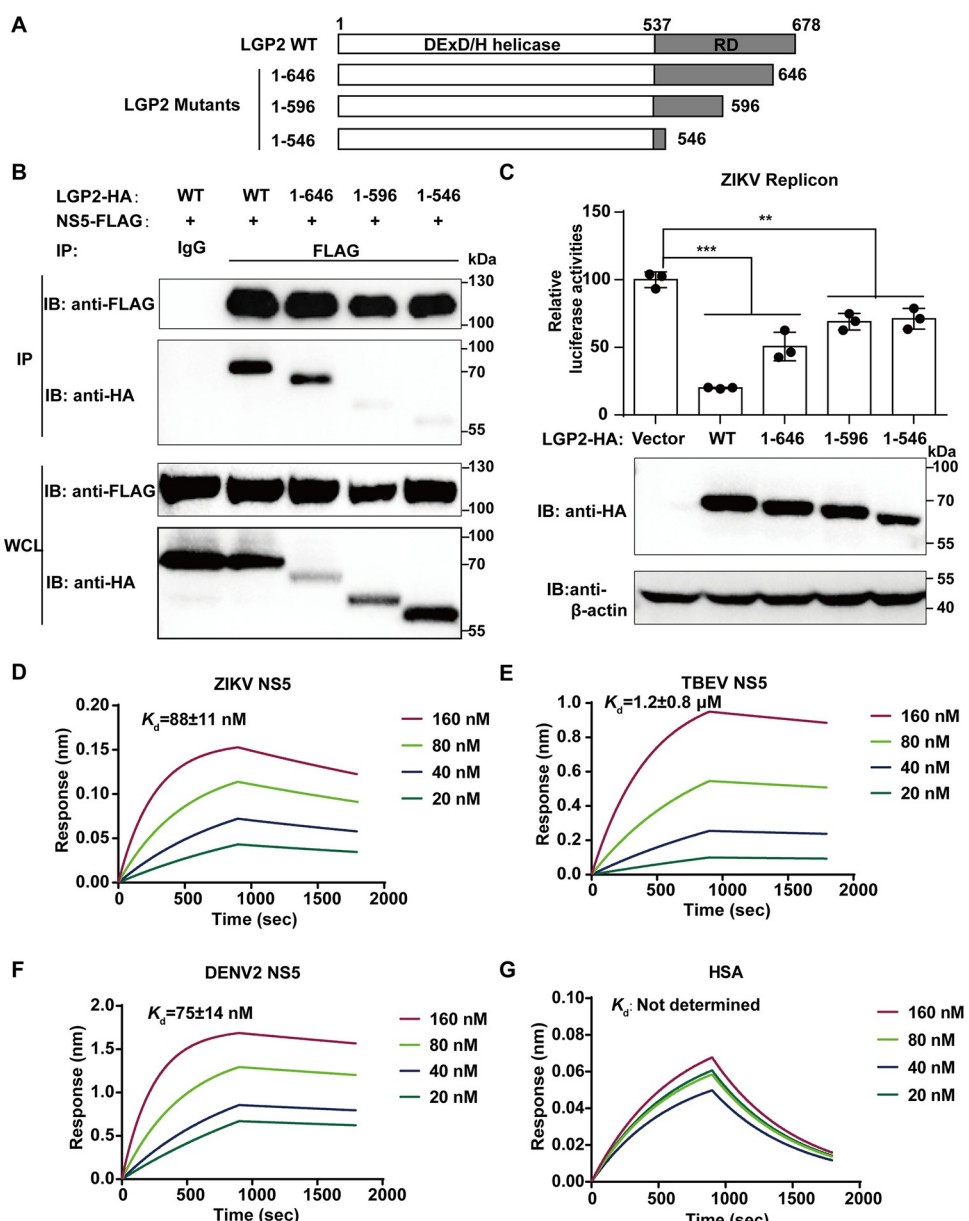

**Fig 5. LGP2 RD directly interacts with NS5.** (A) Schematic diagram of LGP2 truncation mutants. (B) A Co-IP assay of LGP2 or its truncation mutants with ZIKV NS5. Two plasmids were co-transfected into 293T cells and the supernatants of cell lysates were used for Co-IP and Western blot analysis after 24 h. Three independent experiments were performed and images from one experiment were shown. (C) Relative luciferase activities of ZIKV replicon inhibited by LGP2 or its truncation mutants. 400 ng LGP2 WT- or its truncation mutant-overexpressing plasmids were co-transfected with 100 ng viral replicons in 293T cells. Luciferase activities were detected and relative luciferase activities of replicon transfected with vector were set to 100%. Data collected from three independent experiments were shown as Means ± SD (Student's t-test; **: p<0.01, ***: p<0.001). The protein expressions of LGP2 were determined by Western blot. (D-G) Protein binding studies of biotinylated LGP2 RD with ZIKV/TBEV/DENV2 NS5, HSA (negative control), respectively. Biotinylated LGP2 RD protein was diluted (50 μg/mL) with assay buffer. ZIKV/TBEV/DENV2 NS5 and HSA proteins were diluted in a series of concentrations (20, 40, 80, 160 nM). Streptavidin-coated biosensors were used to detect the signal by ForteBio Octet RED system. Data were fitted using Prism. Three independent experiments were performed and results from one representative experiment were shown.

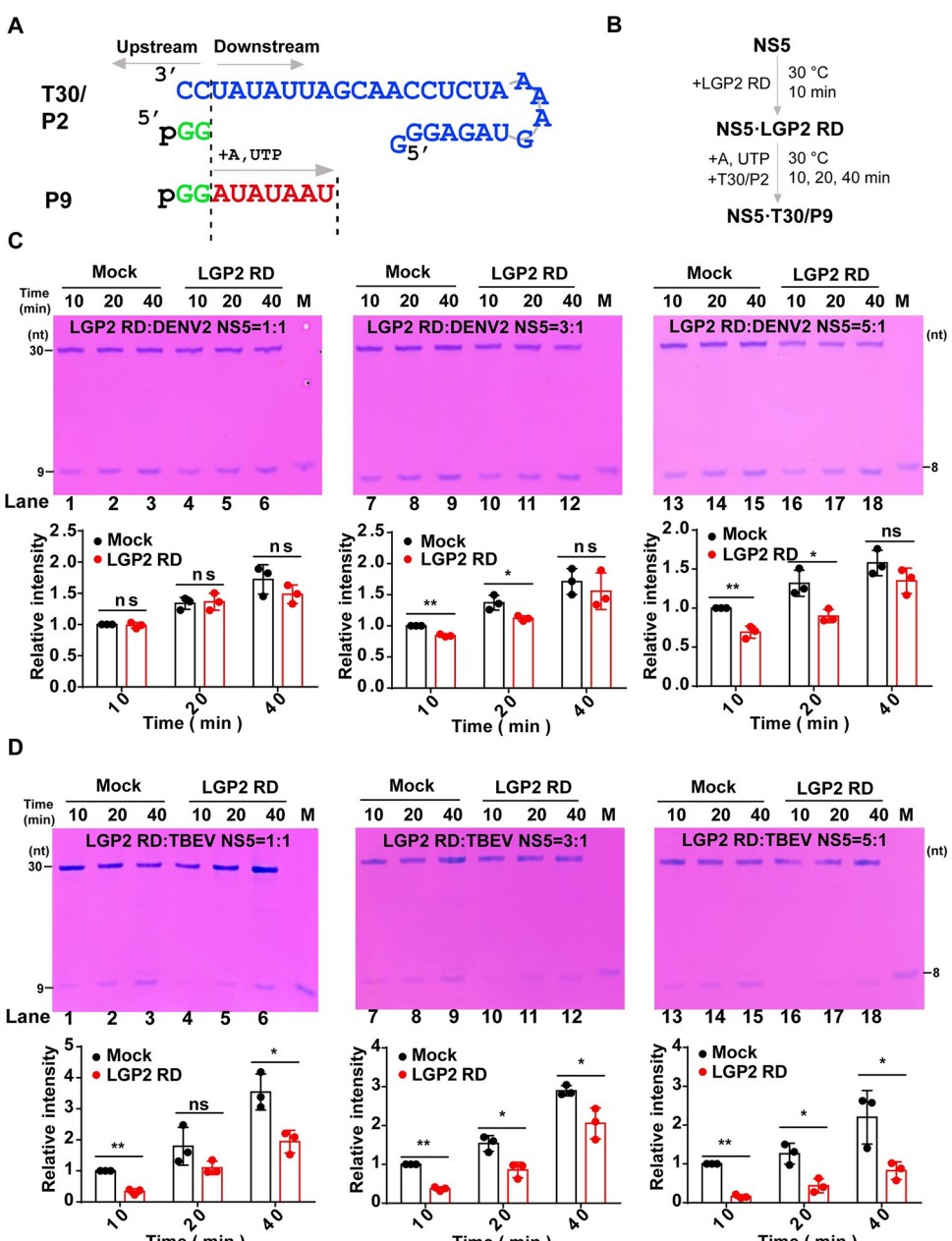

**Fig 6. LGP2 RD inhibits NS5 RdRP activities.** (A) A diagram of T30/P2 RNA construct used in NS5 polymerase assay and the reaction scheme to synthesize a 9-mer product (P9). (B) The reaction flow chart of P2-to-P9 conversion. (C-D) Denaturing polyacrylamide gel electrophoresis (PAGE) analysis of the P2-to-P9 conversion by DENV2 NS5 (C) and TBEV NS5 (D) in the absence (Mock) or presence (LGP2 RD) of LGP2 RD. Top panels: Representative gel images. M: marker, a chemically synthesized 8-mer RNA (P8). This RNA migrates slower than the P9 product bearing a 5′-phosphate as documented in a previous study [46]. Bottom panels: The intensity values were acquired by ImageJ and the relative intensity of P9 at 10 min in the absence of LGP2 RD was set to 1.0. Mean intensities and standard deviations were derived from three independent experiments (Student's t-test; ns: no significant difference, *: p<0.05, **: p<0.01).

P9 product using the T30/P2 construct [46,48]. This P9-containing EC (EC9) is largely insoluble under low-salt condition (e.g., 20 mM NaCl) optimal for RdRP initiation, but becomes much more soluble under high-salt condition (e.g., 200 mM NaCl). Therefore, we removed

ATP and UTP used in EC9 assembly by centrifugation and pellet wash. Then, EC9 in the pellet was resuspended in a high-salt buffer, and CTP was added to allow single-nucleotide addition to synthesizing a 10-mer product (P10) (S6A and S6B Fig). The P9-to-P10 conversion thus allows the probing of polymerase elongation activities. With LGP2 RD provided at a 5:1 molar ratio to NS5, we did not observe inhibitory effect of LGP2 RD in P10 production (S6C Fig). These data therefore suggest that LGP2 RD may not interfere with RdRP elongation process.

We then carried out an assay to study whether LGP2 RD can impair pre-elongation activities of NS5 RdRP. Same as above, LGP2 RD at different molar ratio to NS5 was pre-incubated with NS5 before the only NTP substrate, ATP, was added. A 3-mer product (P3) was synthesized, as directed by the T30/P2 construct (Fig 7A). We performed the *in vitro* pre-elongation activity assay as shown in Fig 7B and observed that the accumulation of P3 products was reduced when LGP2 RD was added for both DENV2 (Fig 7C, compare lanes 1, 3, 7–9, 13–15 to 4, 6, 10–12, 16–18, respectively) and TBEV NS5 (Fig 7D, compare lanes 2–3, 7–8, 14–15 to 5–6, 10–11, 17–18 respectively). ICAM-1 acting as the negative control protein cannot inhibit P3 production by DENV2 NS5 (S5B Fig). Since P2-to-P3 conversion is a multiple turnover process, NS5-RNA rebinding could still occur even if a pre-incubation is carried out. Hence, LGP2 RD impacting on P2-to-P3 conversion cannot be attributed solely to its regulation on RdRP initiation or on NS5-RNA binding.

## The RNA binding capability of LGP2 is not the determinant of its interference with the polymerase pre-elongation activities of NS5

In a previous study, LGP2 RD was shown to bind dsRNA or ssRNA with high affinity [20]. Here we find that LGP2 RD can bind the T30/P2 RNA construct in an electrophoretic mobility shift assay (EMSA) (Fig 8A, lanes 2–4). When LGP2 RD was added to the *in vitro* polymerase assay system, we found that it competitively binds to the RNA and interferes the formation of DENV2 NS5-RNA complex (Fig 8A, compare lanes 6–8 to 5). Therefore, LGP2 RD interference with DENV2 NS5-RNA complex formation may contribute to the inhibitory effect observed in the P2-to-P3 conversion assay.

As LGP2 RD can competitively bind to RNA and interfere the complex formation of NS5 with template RNA, we used an RNA-binding-defective mutant (H576Y, W604A, P606K, K634E, K651E according to a previous study [20], LGP2 RD Mutant) to analyze its inhibitory effect on polymerase activities of DENV2 NS5. Firstly, we verify that LGP2 RD Mutant completely abolishes its RNA binding capability (S7A Fig, lanes 5–7). Furthermore, this mutant does not impair DENV2 NS5-RNA complex formation (Fig 8B, compare lanes 6–8 to 2). Secondly, we used the biolayer interferometry to measure the $K_d$ of LGP2 RD Mutant and DENV2 NS5. The $K_d$ value is 220±57 nM for RD Mutant-DENV2 NS5 interaction (S7B Fig). We then tested this LGP2 RD Mutant in both the P2-to-P9 (Fig 8C) and P2-to-P3 (Fig 8D) conversion assay with a 5:1 molar ratio to DENV2 NS5. The results indicate that the LGP2 RD Mutant has even more obvious inhibition than the LGP2 RD WT (compare lanes 7–9 to 4–6). Finally, we compared their inhibitory effect on virus replication between LGP2 WT and Mutant by co-transfection with ZIKV and TBEV replicon in 293T cells. The results indicate that both proteins have similar inhibitory effect on ZIKV (Fig 8E) and TBEV (Fig 8F) replication. Taken together, our data strongly suggest that LGP2 can downregulate NS5 polymerase pre-elongation activities through direct interaction with NS5 RdRP module.

## Discussion

Flavivirus genome replication occurs in viral RC within ER-associated vesicle packets, and some viral proteins and host factors coordinate to synthesize the viral RNA [49]. NS5, the

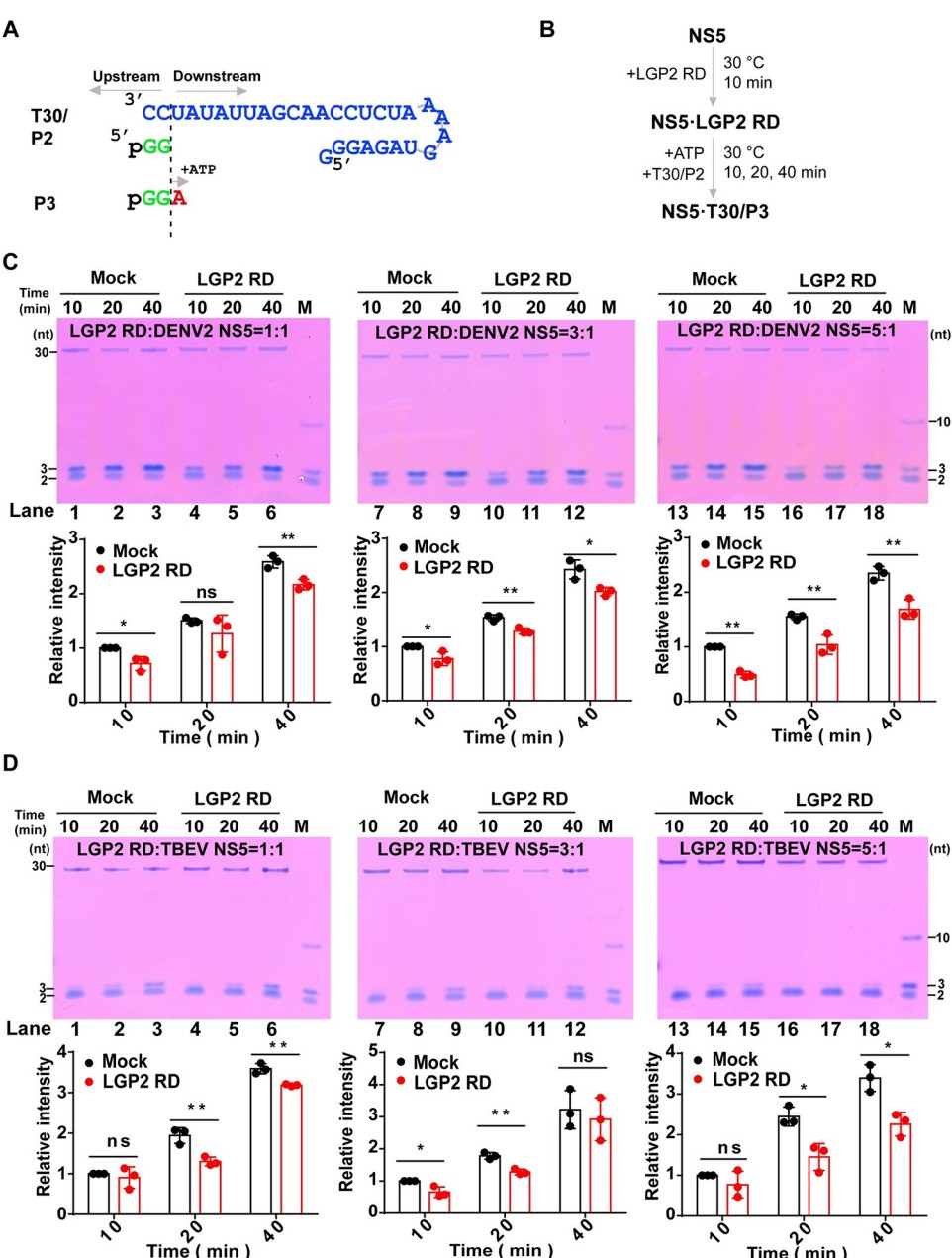

**Fig 7. LGP2 inhibits flavivirus RdRP in pre-elongation stages.** (A) A diagram of construct T30/P2 used in the P2-to-P3 conversion assay and the reaction scheme. (B) The reaction flow chart of P2-to-P3 conversion. (C-D) Denaturing PAGE analysis of the P2-to-P3 conversion by DENV2 NS5 (C) and TBEV NS5 (D) polymerase assays in the absence (Mock) or presence (LGP2 RD) of LGP2 RD. Representative gel images and the band intensity quantitation were shown as in Fig 6. M: marker, a mixture of chemically synthesized 10-mer (5′-hydroxyl), 3-mer (5′-phosphate) and 2-mer (5′-phosphate) RNAs (P10, P3 and P2). The intensity values were acquired by ImageJ and the relative intensity of P3 at 10 min in the absence of LGP2 RD was set to 1.0. Mean intensities and standard deviations were derived from three independent experiments (Student's t-test; ns: no significant difference, *: p<0.05, **: p<0.01).

RdRP, plays a crucial enzymatic role during RNA synthesis as the central RC component. Previous studies indicate that some viral proteins can directly interact with the RdRP to facilitate the RNA synthesis. For instance, ZIKV NS3 interacts with NS5 to stimulate its helicase activity and increase dsRNA intermediate unwinding velocity [43]. Structural studies of the severe

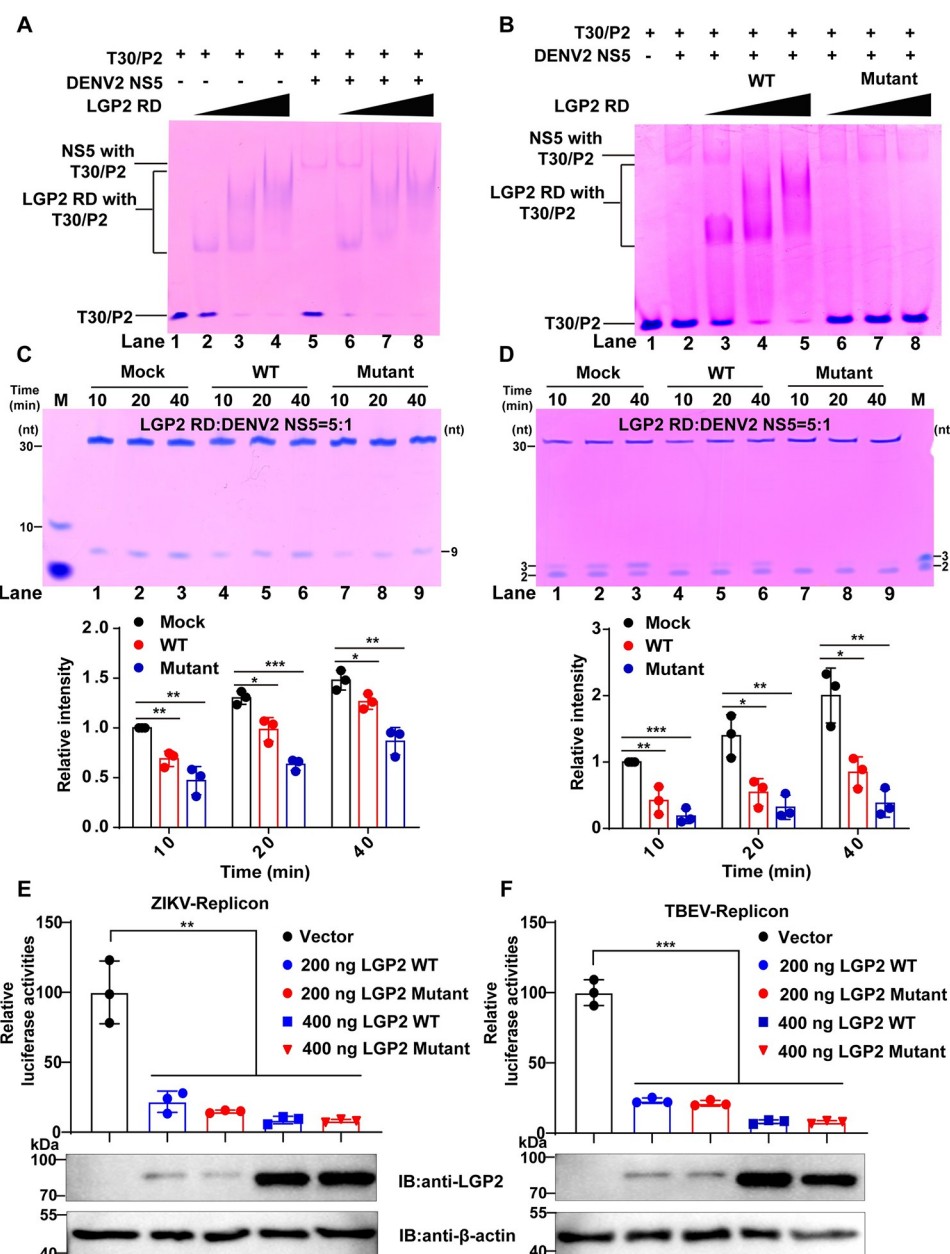

**Fig 8. An RNA-binding-defective mutant of LGP2 RD inhibits NS5 RdRP activity and virus replication.** (A-B) EMSA analysis of LGP2 RD WT (WT) (A) or its mutant (Mutant) (B) binding to RNA in the absence or presence of DENV2 NS5. WT and Mutant were tested at 6, 18, 30 μM concentrations. (C-D) Denaturing PAGE analysis of the P2-to-P9 (C) and P2-to-P3 (D) conversion by DENV2 NS5 in the absence (Mock) or presence of WT or Mutant. Representative gel images and the band intensity quantitation were shown as in Fig 6. Mean intensities and standard deviations were derived from three independent experiments (Student's t-test; *: p<0.05, **: p<0.01, ***: p<0.001). (E-F) Relative luciferase activities of ZIKV (E) and TBEV (F) replicons inhibited by LGP2 mutant. 200 ng or 400 ng LGP2 WT- or its mutant-overexpressing plasmids were co-transfected with 100 ng viral replicons in 293T cells. Luciferase activities were detected, and relative luciferase activities of replicon transfected with vector were set to 100%. Data collected from three independent experiments were shown as Means ± SD (Student's t-test; **: p<0.01, ***: p<0.001).

acute respiratory syndrome coronavirus 2 (SARS-CoV-2) RdRP complexes suggest that the RdRP activities of nsp12 need the assistance of nsp7 and nsp8 as cofactors [50]. In our study, competitive Co-IP assay data between LGP2 and NS3 indicate that LGP2 affects binding between NS3 and NS5. LGP2 outcompeting with NS3 in binding to NS5 in the RC may further downregulate viral genome replication. By contrast, DExD/H box RNA helicase, DDX17, facilitates human-adapted highly pathogenic avian influenza A (HPAI) viruses of the H5N1 polymerase adaptation in the RC and viral RNA synthesis in human cells [51], and porcine reproductive and respiratory syndrome virus (PRRSV) nucleocapsid (N) protein recruits DExD/H box RNA helicase, DHX9, into the RC to promote viral replication [52]. Different from DDX17 and DHX9 that play auxiliary roles, LGP2 may act as a host restriction factor to modulate flavivirus genome replication.

In addition, some host factors interact with other components in viral RC and regulate viral RNA synthesis by RdRP. One of the best-described host factors involved in influenza virus replication, RNA polymerase II (Pol II), interacts with viral RC to provide access to capped RNA molecules for cap snatching and specifically promote viral transcription [53]. rRNA processing 1 homolog B (RRP1B) interacts with influenza virus RC and facilitate the binding of RdRP to capped host mRNAs [54]. Some chaperonins, heat shock proteins (HSP90, HSP40) [55,56] and cytosolic chaperonin containing TCP-1 (CCT) [57] are reported that they interact with PA and PB2 of the RC and enhance viral replication. Taken together, the mechanism of flavivirus genome replication downregulation by LGP2 through direct interaction with RdRP is different from the aforementioned host protein-related mechanisms that mostly enhance genome replication, through interactions with RdRP or other components within the RC. We have identified a distinct antiviral regulation for LGP2 completely different from the previous immune regulation of viral infection by regulating RLRs signaling pathway [19], affecting ubiquitination [39], protein interaction [58], and protein cleavage [30].

Flavivirus NS5 not only plays key roles in viral genome replication and transcription, but also participates in other cellular events beyond the RC compartments. Many host factors that directly interact with viral RdRP have been identified to date. A few of them act as a modulator for an innate immune response during virus infection. Flavivirus NS5 interacts with multiple critical protein in the JAK-STAT signalling, such as prolidase (PEPD) [59], the PDZ protein scribble (hScrib) [60], and STAT2. STAT2 binds to ZIKV and DENV2 NS5, and blocks its phosphorylation and causes its degradation [61]. It was reported that ZIKV NS5 interacts with I-kappa-B kinase epsilon (IKKε) and TANK-binding kinase 1 (TBK1) regulated IFN production by inhibiting phosphorylation of Interferon regulatory factor 3 (IRF3) [62,63]. These host factors are signal transduction molecules at the downstream of RLRs. Besides, TBEV NS5 RdRP activates IFN signalling dependent on RIG-I and MDA5 which are the upstream receptor of IFN signalling [64]. Whether ZIKV and TBEV RdRP binding to LGP2 interferes its regulation for RIG-I and MDA5 is still unclear. It was reported that ZIKV NS5 binds to CARD domain of RIG-I and inhibits the production of IFNβ [65]. Because of lacking the CARD domain, ZIKV NS5 presumably interacts with LGP2 in a different way. The LGP2 RD share only 29% and 34% amino acid identities with RIG-I and MDA5 RD, respectively [21]. Therefore, RD domain of LGP2 could interact with NS5 in a unique way when regulating RdRP's function.

In summary, we find a distinct antiviral regulation mechanism of LGP2 that flavivirus RdRP pre-elongation activities were impaired by LGP2 through its directly interaction with RdRP module. Our work thus provides a basis for further understanding the working and regulation mechanisms of flavivirus RC, and in particular for the contribution of host proteins in viral genome replication and transcription.

## Materials and methods

### Cells and viruses

African green monkey kidney epithelial cell line (Vero, ATCC CCL-81), human non-small cell lung cancer cell line (A549, ATCC-CCL-185), HeLa cell line (ATCC-CRM-CCL-2), and human embryonic kidney cell line (HEK293T, ATCC-CRL-3216) were cultured in Dulbecco's Modified Eagle Medium (DMEM, Invitrogen, USA). Human astrocytoma cells (CCF-STTG1, ATCC-CRL-1718) were cultured in RPMI 1640 Medium (Gibco, USA). These cells were maintained in the medium containing 10% (v/v) fetal bovine serum (Life Technology, Australia) in 5% (v/v) $CO_2$ at 37˚C. *Aedes albopictus* clone C6/36 cell line (ATCC-CRL-1660) was cultured in RPMI 1640 Medium containing 10% (v/v) FBS in 5% (v/v) $CO_2$ at 28˚C. ZIKV strain (Zika virus/SZ01/2016/China, GenBank: KU866423.2) was obtained from Wuhan Institute of Virology, Chinese Academy Sciences [66].

### Plasmid construction

Sequences of ZIKV and TBEV non-structural proteins and *lgp2* were amplified from cDNA using gene-specific primers. Then, they were cloned into mammalian expression vector pCAGGS with N-terminus FLAG or HA tag at the *Eco*RI and *Xho*I sites. *Lgp2-flag*, *mCherry-flag*, and *Lgp2-bir A\** gene sequences were obtained using PCR and cloned into lentiviral packaging plasmid pCDH-CMV-MCS-EF1-CopGFP-T2A-puro at the *Eco*RI and *Not*I sites. Site-directed mutagenesis sequences of *lgp2*, *lgp2-gfp* gene sequences, and truncation mutant sequences of *ns5* and *lgp2* were obtained using overlapping PCR amplified from WT plasmids and cloned into mammalian expression vector pCAGGS with N-terminus FLAG or HA tag at the *Eco*RI and *Xho*I sites. ZIKV and TBEV replicons were constructed by using overlapping PCR amplified from cDNA. PCR fragments including N-terminus 22 amino acids of capsid protein, Rluc containing FMDV 2A, C-terminus 30 amino acids of envelope protein, and non-structural protein were cloned into the low-copy-number plasmid pACYC177 at the *Kpn*I and *Xho*I sites (Fig 2A).

### LGP2 siRNA knockdown and overexpression

Three target sequences for human *lgp2* were si-1: GAGCCAAGGTGGTTGTATT, si-2: GCCAGTACCTAGAACTTAA, si-3: GCAATGTGGTGGTGCGTTA (RIBOBIO, China). Non-targeting siRNA (si-nc) was used as a negative control for human *lgp2* siRNA. CCF-STTG1 cells were plated into 12-well plates at $5\times10^5$ cells/well. SiRNAs were transfected by Lipofectamine 3000 (Invitrogen, USA), and after 16–24 h the LGP2-knockdown CCF-STTG1 cells were infected by ZIKV (MOI = 0.1). The RT-qPCR, Western blot and virus titer samples were obtained at 12, 24, and 36 hpi. We utilized the third-generation lentivirus packaging system to package lentiviruses in 293T cell line. The lentivirus-containing cell supernatants were collected, and then used to infect CCF-STTG1 cells. At 48 hpi, 1640 medium with 10% (v/v) FBS and 1 μg/mL puromycin was used to culture and screen the positive cells. Stably LGP2- and mCherry-overexpressing CCF-STTG1 cell lines were established and detected by Western blot. CCF-STTG1 cells were plated into 12-well plates at $5\times10^5$ cells/well, and infected by ZIKV (MOI = 0.1). The RT-qPCR and Western blot samples were obtained at 12, 24, and 36 hpi.

### RT-qPCR

Total RNA was extracted from cells by using TRIZOL Reagent (Invitrogen, USA) following the manufacturer's protocol. The first-strand cDNA was reverse transcribed from 2 μg total

RNA using FastKing gDNA Dispelling RT Supermix (TIANGEN, China). The 20 μL of quantitative PCR reaction mix includes the first-strand cDNA, quantitative primers, and iTaq Universal SYBR Green Supermix (Bio-Rad, USA). PCR amplified procedure involves activation at 95°C for 3 min and 40 amplification cycles of 95°C for 5 s and 60°C for 45 s in a Bio-Rad CFX real-time quantitative PCR system. The quantitative primers are shown in S1 Table [29,38]. The genes' relative expression were analyzed using $2^{-\triangle\triangle ct}$ method.

## Western blot

Total protein was separated by SDS-PAGE using 10% or 12% (w/v) polyacrylamide gels. Then, the separated protein was transferred onto 0.45 μm Immobilon-P polyvinylidene fluoride membranes (Millipore, USA) in transfer buffer (30 mM Tris, 200 mM glycine, 20% (v/v) methanol) for 150 min at 4°C at 90 V. Immunoblot membranes were blocked using 5% (w/v) BSA albumin fraction V (Biofroxx, Germany) dissolved in TBS-T for 1 h at 37°C. Then, immunoblot membranes were incubated with primary antibodies and secondary antibodies conjugated with HRP. After washing with TBS-T, immunoblot membranes were visualized and analyzed using a Bio-Rad imaging system with an Immobilon Eastern Chemiluminescent HRP substrate (Millipore, USA). The bands were analyzed by using Image Lab 4.0.1 and ImageJ.

The primary antibodies used in this study were as follows: LGP2 (Abcam, ab67270), HA (Cell Signaling Technology, #3724), FLAG (Sigma, F1804), β-actin (Beyotime, AF0003), HRP-streptavidin (Beyotime, A0303), ZIKV NS5 (GeneTex, GTX133312-S), ZIKV NS3 (GeneTex, GTX133320), ZIKV envelope protein (ENV) (BioFront Technologies, BF-1176-56). All antibodies were diluted following the manufacturer's protocol.

## Plaque assay

Vero cells at 80% confluence were planted into 24-well plates, and were inoculated with 100 μL of 10-fold serial dilution of the virus samples in serum-free DMEM. After 1.5 h incubation, 1 mL culture of 1.25% (w/v) methylcellulose-containing 2% FBS (v/v) was added to each well. After incubation for 4 days, cells were fixed with 4% (w/v) paraformaldehyde and stained with 0.5% (w/v) crystal violet. Plaque numbers were recorded after rinsing the plates with deionized water.

## Co-immunoprecipitation assay (Co-IP)

The plasmids co-transfected 293T cells or plasmid transfected and then ZIKV infected A549 cells were harvested at 24 or 48 hpi and lysed using lysis buffer (Beyotime, China) supplemented with phenylmethanesulfonyl fluoride (PMSF) (Beyotime, China) and protease inhibitor cocktail tablets (Roche, Switzerland) following the manufacturer's instructions. After centrifugation, the supernatants were incubated with FLAG antibodies (Sigma, F1804) and protein G magnetic beads (Invitrogen, USA) for 60 min at room temperature (r.t.). Then washed using PBS-T buffer five times, the immunoprecipitates were eluted by boiling for 10 min in SDS-PAGE loading buffer. For digesting RNA, the supernatants of cell lysates were pre-treated by RNase A (4, 10 μg/mL) and RNase I (100, 200 U/mL) for 4 h at 4°C [45]. Finally, the protein samples were analyzed by using Western blot.

## Luciferase reporter assay

293T or Vero cells were planted into 24-well plates, and co-transfected with LGP2-overexpressing plasmids and viral replicons, or LGP2-overexpressing plasmids, reporter plasmid p-IFNβ-Luc and control plasmid pRL-TK after being cultured for 16–20 h. Vector plasmid

(pCAGGS-HA) was added to ensure the equally total amount of transfected plasmid DNA in each sample. For examining the activity of IFNβ suppressed by LGP2, cells were infected with 100HA SEV. After 24 h of incubation, the medium was removed, and the cells were washed again with PBS. 100 μL assay lysis buffer was added to each well, and the culture plates were rocked at r. t. for 15 min. Then, cell lysate samples were transferred to tubes and centrifuged for 30 s at 16000 RCF at 4˚C. According to its technical manual, the supernatants of cell lysates were detected using *Renilla* luciferase assay system (Promega, USA) or Dual-Luciferase Reporter assay system (Promega, USA), and the expression of LGP2 was analyzed using Western blot.

## Confocal immunofluorescence microscopy

HeLa cells were planted into glass slides with $5 \times 10^4$ cells per dish. Then, LGP2-GFP-overexpressing plasmids were transfected by using Lipofectamine 3000. After 16–20 h of incubation, ZIKV infected these cells at MOI = 5. At 24 hpi, the slides were washed with PBS, fixed with 4% (w/v) paraformaldehyde, and then permeabilized with 0.2% (v/v) Triton X-100 for 12 min. Cells were blocked by using 2% (w/v) BSA and 3% (v/v) normal goat serum (NGS) (Biofly, China) dissolved in PBS for 1 h at 37˚C. Cells were washed twice and then incubated in primary antibodies (1:200) diluted with PBS containing 3% (v/v) NGS overnight at 4˚C. After washing three times, cells were incubated with anti-rabbit IgG (H+L) F(ab')$_2$ fragment (Alexa Fluor 555) (Cell signaling technology, #4413S) (1:500) and Goat anti-Mouse IgG (H+L) Cross-Adsorbed Secondary Antibody (Alexa Fluor 633) (Invitrogen, A21050) (1:1000) diluted with PBS containing 3% (v/v) NGS for 1 h at r.t. After washing five times, cell nuclei were dyed with Hoechst 33258 (Beyotime, C1018) for 8 min at r.t. Imaging was performed using a Leica STEL-LARIS 8 WILL microscope (DMI8) based on a 60× oil-immersion objective as well as 405/488/555/633 nm solid-state lasers. Image processing and colocalization analysis was performed in LAS X and Imaris software. Background correction and MCCs (using the Coloc 2 plugin) were obtained by ImageJ. The primary antibodies used in this study were as follows: ZIKV NS5 (GeneTex, GTX133312-S); dsRNA (Nordic-MUbio, 10010500).

## Affinity capture of biotinylated proteins

Humanized *birA* gene with R118G mutation (BirA*) was synthesized at Wuhan GeneCreate Biological Engineering (China). The sequence of LGP2-BirA* fusion protein was cloned into lentiviral packaging plasmid pCDH-CMV-MCS-EF1-CopGFP-T2A-puro. We utilize the third-generation lentivirus packaging system to package lentiviruses in 293T cell line. The lentivirus-containing cell supernatants were collected, and infected CCF-STTG1 cells. At 48 hpi, 1640 medium with 10% (v/v) FBS and 1 μg/mL puromycin was used to culture and screen the positive cells. Stably LGP2-BirA*-overexpressing CCF-STTG1 cell line was established and detected by Western blot. Then, these cells were planted into 60 mm cell culture dishes. After cultured for 24 h, these cells were infected by ZIKV at MOI = 0.1 and incubated in complete media supplemented with 1 μg/mL puromycin and 50 μM biotin for 48 h. These cells were washed by PBS and lysed in cell lysis buffer. The supernatants of cell lysates were collected by centrifugation and incubated overnight with 100 μL streptavidin magnetic beads (Biolinkedin, China). The beads were collected and washed as described in a previous study [42]. The immunoprecipitates were eluted by boiling for 10 min in SDS-PAGE loading buffer. Finally, the protein samples were analyzed by using Western blot.

## Protein production and purification

WT and RNA-binding-defective mutant proteins of LGP2 RD (residues 537–678) were cloned into a pET28a expression vector and expressed in *E. coli* Rosetta DE3. *E. coli* culture, and

protein expression conditions, purification methods, and buffers were described in a previous study [20]. The LGP2 RD proteins were stored in GF buffer (30 mM Tris-HCl [pH 7.5], 400 mM NaCl, 2 mM dithiothreitol (DTT), 10 μM ZnCl$_2$, 10% (v/v) glycerol) at -80°C after being flash-frozen. NS5 proteins of DENV2 and TBEV were expressed and purified according to previously described procedures [46,47]. The expression and purification procedures of ZIKV NS5 protein were the same as those of DENV2 NS5.

## Biolayer interferometry assay

The protein binding study was carried out using the ForteBio Octet RED system and performed in a 96-well plate at r.t. with shaking at 1000 rpm. The assay buffer was PBS-T containing 0.1% (w/v) BSA filtered by using a 0.22 μm filter. The protein buffer of LGP2 RD and NS5 was replaced with assay buffer using a protein ultrafiltration tube. Biotin (10 mg/mL) was added into LGP2 RD protein at 3:1 molar ratio to biotinylate the LGP2 RD protein at r.t. for 1 h. Then, the buffer was once again replaced with assay buffer to remove the free biotin. Biotinylated LGP2 RD protein was diluted (50 μg/mL) with assay buffer. NS5 and human serum albumin (HSA) (Acmec, HSA-A93920) proteins were diluted to a series of concentrations (20, 40, 80, 160 nM), and added into 96-well plate (200 μL/well). Streptavidin-coated biosensors were washed with assay for buffer 120 s and loaded with biotinylated LGP2 RD protein for 600 s, followed by 120 s washing. Then biosensors were reacted with NS5 protein for 900 s association and 900 s dissociation. The data were fitted using Prism.

## *In vitro* RdRP assays

For the *in vitro* RdRP assay monitoring P2-to-P9 conversion, a 20 μL reaction mixture containing 6 μM NS5, 4 μM RNA construct T30/P2 (4 μM T30 and 80 μM P2), 300 μM ATP, and 300 μM UTP in corresponding reaction buffer (Buffer A for DENV2 NS5 and Buffer B for TBEV NS5) was incubated at 30°C for 10, 20, and 40 min before being quenched by an equal volume of 2×stop solution (95% (v/v) formamide, 20 mM EDTA (pH 8.0), 0.02% (w/v) bromphenol blue). For the assay monitoring P2-to-P3 conversion, the reaction procedures were the same as above except that UTP was omitted from the mixture. For the single-nucleotide elongation assay, P2-to-P9 conversion was carried out as described above with ATP and UTP concentrations both elevated to 600 μM, and the reaction mixture was incubated at 30°C for 60 min. The reaction mixture was then centrifuged at 16000 g at 4°C for 5 min. The supernatant was removed, and pellet was subjected to gentle washes twice with buffer A before resuspended in a high-salt Buffer C. LGP2 RD was then added to the resuspension solution at a 5:1 molar ratio to NS5 and the mixture was incubated at 30°C for 10 min before chilled on ice. CTP was added to a final concentration of 300 μM with the reaction mixture still on ice, and the mixture was incubated at 30°C for 0, 1, 2, 5, 30 min (S6B Fig) before being quenched by an equal volume of 2×stop solution. Buffer A: 50 mM Tris (pH 7.5), 20 mM NaCl, 5 mM MgCl$_2$, 5 mM DTT. Buffer B: 50 mM Tris (pH 7.5), 20 mM NaCl, 5 mM MnCl$_2$, 2 mM MgCl$_2$, 5 mM TCEP, 0.01% (v/v) *n*-dodecyl-β-D-maltoside (DDM). Buffer C: 50 mM Tris (pH 7.5), 200 mM NaCl, 5 mM MgCl$_2$, 5 mM DTT.

To assess LGP2 RD impact on P2-to-P3 or P2-to-P9 conversion, NS5 and various amount of LGP2 RD were pre-incubated at 30°C for 10 min. In parallel experiments, ICAM-1 (Sino Biological, 10346-HCCH-50) belonging to a subset of Ig-like superfamily proteins replaced LGP2 RD as the negative control protein. Unless otherwise indicated, the procedures for PAGE, gel staining, and quantitation were the same as described in a previously study [67].

## Electrophoretic mobility shift assay (EMSA)

4 μM T30/P2 was incubated with various concentrations (6, 18, 30 μM) of LGP2 RD or its mutant and DENV2 NS5 (6 μM) in a modified Buffer A (The NaCl concentration was elevated to 89 mM) at 30˚C for 10 min. 8% (w/v) native polyacrylamide gels were used to resolve the RNA and RNA-protein complex. Gel staining and image acquisition were the same as in a previously study [67].

## Statistical analysis

Wherever applicable, data are presented as Mean ± SD, and Student's *t*-test was used for all statistical analyses. Differences were considered significant when the p-value was less than 0.05.

## Supporting information

**S1 Fig. ZIKV infection upregulates the expression of LGP2 in CCF-STTG1 cells.** (A-D) CCF-STTG1 cells were infected by ZIKV (MOI = 0.1). RT-qPCR and Western blot samples were obtained at 12, 24, 36, and 48 hpi. (A) The protein expressions of LGP2, RIG-I, MDA5, ZIKV ENV, and β-actin determined by Western blot. (B-D) The relative mRNA levels of LGP2 (B), MDA5 (C), and RIG-I (D) quantified by RT-qPCR. Data collected from three independent experiments are shown as Means ± SD (Student's t-test; ***: $p < 0.001$).
(TIF)

**S2 Fig. LGP2 knockdown increases the transcript levels of the IFN signaling and proinflammatory cytokines during ZIKV infection.** (A-H) The LGP2-knockdown CCF-STTG1 cells were infected by ZIKV (MOI = 0.1), and RT-qPCR samples were obtained at 0, 12, 24 and 36 hpi. The relative mRNA levels of IFNβ (A), IκBα (B), ISG56 (C), IL6 (D), Mx1 (E), and CXCL10 (F) were quantified by RT-qPCR. Mock, CCF-STTG1 cells without any treatment. Si-nc and si-2, CCF-STTG1 cells were transfected with siRNAs (si-nc and si-2, respectively). Data collected from three independent experiments are shown as Means ± SD (Student's t-test; *: $p < 0.05$, **: $p < 0.01$, ***: $p < 0.001$).
(TIF)

**S3 Fig. LGP2 overexpression decreases the transcript levels of the IFN signaling and proinflammatory cytokines during ZIKV infection.** (A-H) The stably mCherry- and LGP2-overexpressing CCF-STTG1 cell lines were infected by ZIKV (MOI = 0.1), and RT-qPCR samples were obtained at 0, 12, 24 and 36 hpi. The relative mRNA levels of IFNβ (A), IκBα (B), ISG56 (C), IL6 (D), Mx1 (E), and CXCL10 (F) were quantified by RT-qPCR. Mock, CCF-STTG1 cells without any treatment. MCherry and LGP2, mCherry- and LGP2-overexpressing CCF-STTG1 cell lines. Data collected from three independent experiments are shown as Means ± SD (Student's t-test; *: $p < 0.05$, **: $p < 0.01$, ***: $p < 0.001$).
(TIF)

**S4 Fig. LGP2 negatively regulates the IFN induction.** Fold change luciferase activities of IFNβ. LGP2 WT or its mutants (MIV and MV) were co-transfected with reporter plasmid p-IFNβ-Luc, and control plasmid pRL-TK in 293T cells. After 24 h of transfection, cells were infected with 100HA SEV. The supernatants of cell lysates were obtained and detected by using a Dual luciferase assay system and Western blot at 24 hpi. Fold change luciferase activities represent the Fluc/Rluc ratio ($\times 10^4$). Data collected from three independent experiments are shown as Means ± SD (Student's t-test; ***: $p < 0.001$).
(TIF)

**S5 Fig. ICAM-1 does not inhibit NS5 RdRP activities.** (A-B) Denaturing PAGE analysis of the P2-to-P9 (A) and P2-to-P3 (B) conversion by DENV2 NS5 in the absence (Mock) or presence (ICAM-1) of ICAM-1. Left panels: Representative gel images. M: marker, a mixture of chemically synthesized 10-mer (5′-hydroxyl), 3-mer (5′-phosphate) and 2-mer (5′-phosphate) RNAs (P10, P3 and P2). Right panels: The intensities of P9 or P3 bands were analyzed by ImageJ. The relative intensity at 10 min in the absence of ICAM-1 was set to 1.0.
(TIF)

**S6 Fig. LGP2 RD does not affect flavivirus RdRP in the elongation stage.** (A) A diagram of T30/P2 RNA construct used in NS5 polymerase assay and the reaction scheme to synthesize a 10-mer product (P10). (B) The reaction flow chart of P9-to-P10 conversion. (C) Denaturing PAGE analysis of the P9-to-P10 conversion by DENV2 NS5 in the presence of $H_2O$ (Mock), GF buffer of LGP2 RD (Buffer) and LGP2 RD with GF buffer (LGP2 RD). Top panels: Representative gel images. M: marker, a chemically synthesized 10-mer (5′-hydroxyl) RNA (P10). This RNA migrates slower than the P10 product bearing a 5′-phosphate as documented in a previous study [46]. Bottom panels: The intensities of P10 bands were analyzed by ImageJ. The relative intensity at 0 min was set to 1.0.
(TIF)

**S7 Fig. RNA-binding-defective mutant of LGP2 RD abolishes RNA binding, but can still interact with NS5.** (A) EMSA analysis of LGP2 RD WT (WT) or its mutant (Mutant) binding to RNA. WT and Mutant were tested at 6, 18, 30 μM concentrations. (B) Detection of NS5-LGP2 binding. Interactions between biotinylated Mutant and DENV2 NS5 was analyzed by biolayer interferometry. DENV2 NS5 was diluted in a series of concentrations (20, 40, 80, 160 nM).
(TIF)

**S1 Table. Primers used for RT-qPCR assay.**
(PDF)

## Acknowledgments

We thank Dr. Bo Zhang for providing the ZIKV NS5 gene, Dr. Xi Zhou for providing the FLAG-tagged Dicer plasmid, and Dr. Ding Gao at the Institutional Center for Shared Technologies and Facilities of Wuhan Institute of Virology, Chinese Academy of Sciences, for assistance in confocal immunofluorescence microscopy and biolayer interferometry assays.

## Author Contributions

**Conceptualization:** Zhenhua Zheng, Hanzhong Wang, Jianping Tao, Peng Gong.

**Data curation:** Zhongyuan Tan, Jiqin Wu, Zhenhua Zheng, Hanzhong Wang, Jianping Tao, Peng Gong.

**Formal analysis:** Zhongyuan Tan, Jiqin Wu, Li Huang, Ting Wang, Zhenhua Zheng, Jianhui Zhang, Xianliang Ke, Yuan Zhang, Yan Liu, Hanzhong Wang, Jianping Tao, Peng Gong.

**Funding acquisition:** Zhongyuan Tan, Jiqin Wu, Zhenhua Zheng, Peng Gong.

**Investigation:** Zhongyuan Tan, Jiqin Wu.

**Methodology:** Zhongyuan Tan, Jiqin Wu, Li Huang, Ting Wang, Zhenhua Zheng, Jianhui Zhang, Xianliang Ke, Yuan Zhang, Yan Liu.

**Project administration:** Zhongyuan Tan, Jiqin Wu, Zhenhua Zheng, Yan Liu, Hanzhong Wang, Jianping Tao, Peng Gong.

**Resources:** Zhenhua Zheng, Hanzhong Wang, Jianping Tao, Peng Gong.

**Supervision:** Zhenhua Zheng, Hanzhong Wang, Jianping Tao, Peng Gong.

**Visualization:** Zhongyuan Tan, Jiqin Wu.

**Writing – original draft:** Zhongyuan Tan, Jiqin Wu, Peng Gong.

**Writing – review & editing:** Zhongyuan Tan, Jiqin Wu, Li Huang, Ting Wang, Zhenhua Zheng, Jianhui Zhang, Xianliang Ke, Yuan Zhang, Yan Liu, Hanzhong Wang, Jianping Tao, Peng Gong.

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
