## [Decision Letter · Decision Letter 0]

5 Mar 2023

Dear Dr. Gong,

Thank you very much for submitting your manuscript "LGP2 directly interacts with flavivirus NS5 RNA-dependent RNA polymerase and downregulates its pre-elongation activities" for consideration at PLOS Pathogens. As with all papers reviewed by the journal, your manuscript was reviewed by members of the editorial board and by several independent reviewers. In light of the reviews (below this email), we would like to invite the resubmission of a significantly-revised version that takes into account the reviewers' comments.

We cannot make any decision about publication until we have seen the revised manuscript and your response to the reviewers' comments. Your revised manuscript is also likely to be sent to reviewers for further evaluation.

Sincerely,

Alison Kell

Guest Editor

PLOS Pathogens

Sonja Best

Section Editor

PLOS Pathogens

Kasturi Haldar

Editor-in-Chief

PLOS Pathogens

orcid.org/0000-0001-5065-158X

Michael Malim

Editor-in-Chief

PLOS Pathogens

orcid.org/0000-0002-7699-2064

Reviewer's Responses to Questions

**Part I - Summary**

Reviewer #1: In the paper,” LGP2 directly interacts with flavivirus NS5 RNA-dependent RNA polymerase and downregulates its pre-elongation activities” by Zhongyuan Tan et al. authors investigate the antiviral activities of LGP2 towards flaviviruses. LGP2 is a RIG-I-like receptor dsRNA helicase enzyme which has been shown to be essential for producing effective antiviral responses against viruses recognized by RIG-I and MDA5. However, LGP2 lacks the CARD domain that RIG-I and MDA5 have and therefor the role of LGP2 is more like a regulator than a direct trigger of the downstream signalling pathways. LGP2 has been reported to regulate RIG-I and MDA5 both negatively and positively during viral infection but the protein has also been shown to interact directly with both host and viral proteins. In the current paper by Zhongyuan Tan et al. authors characterize the role of LGP2 during flavivirus infection and find that the protein directly interacts with viral RdRP NS5 through its regulatory domain (RD) and thereby inhibits the activity of NS5 during the pre-elongation step. The overall findings are of interest however, the manuscript is lacking information how the experiments were performed and statistics, thus it is not possible to follow and judge the quality. The whole manuscript would benefit if all data points in the bar graphs are visible. Some of the statements made in the text are too strong and not supported by the data provided.

Reviewer #2: In the manuscript Tan et al. describe the inhibitory effect on flavivirus replication of LGP2 RD binding to the NS5 RdRP domain. As a whole, the results are interesting and may provide evidence of new antiviral mechanisms operating in infected cells alternative or complementary to IFN-based response. The interaction of NS5 with LGP2 RD seems convincing but the IFN-independent inhibitory effect on replication based on KD and polymerase assays needs to be better discussed and some experiments refined.

Reviewer #3: This manuscript discusses the role of LGP2 in modulating flavivirus replication. The replication of the flavivirus occurs in the viral replication complex within ER-associated vesicle packets, where viral proteins and host factors coordinate to synthesize the viral RNA. NS5, the RdRP, plays a crucial role during RNA synthesis as the central RC component. The manuscript explains that LGP2 can affect the binding between NS3 and NS5, which can further downregulate viral genome replication, suggesting that LGP2 may act as a host restriction factor to modulate flavivirus replication. Overall, the manuscript provides a comprehensive overview of LGP2's role in modulating flavivirus replication, providing valuable insights into the mechanisms of viral replication and host-virus interactions. The manuscript is well-structured and well-written, with appropriate references to support the claims made by the authors. However, there are issues that need to be addressed before reaching such conclusion.

**Part II – Major Issues: Key Experiments Required for Acceptance**

Reviewer #1: For figure 3C the data showing colocalization is very weak. To state that two proteins are colocalizing statistical methods looking into many cells need to be used. If one protein is all over the cytoplasm that does not mean they are colocalizing. The authors refer to a previously described method however the method is not described in full in that paper, that information needs to be included in this manuscript. The Mander’s coefficient between dsRNA and LGP2 could for example be used looking at individual cells in a sample to verify that LGP2 is around replication complexes. The statement on line 216 that “these findings demonstrate the involvement of LGP2 in the RC” based on fig 3 is way to strong and needs to be rephrased.

For figure 2 it is not clear how this experiment was performed, in either material and method or figure legend. It seems that the total amount of plasmid DNA transfected into cells were different between samples, this would affect the transfection efficiency and the replication of the replicon making the results difficult to interpret.

Please explain the statistical analysis performed for si-3 in panel C of figure 1 as both are shown to be statistically significant. The data is also contradictory that si-3 in the 48 h timepoint shows no effect on ZIKV RNA but highest level of titres panels C and D. It is misleading to do grouped statistics for the three siRNAs in all graphs but the 24h PFU/ml, please correct this.

Why is the level of infection different between 1C and 1E since the MOI is the same? And why is the infection so low at 48h in 1E compared to 1C? Please discuss this in the text.

Although there is a clear difference in RNA level in Fig 1E at 24 and 36 h the F panel is not showing convincingly that LGP2 expression decreases E protein in the cell. Please quantify the blots and show in a graph with all the data point visual. Also the triangles needs to be shorter as lane 1, 4, 7 and 10 does not contain the LGP2.

In figure 5 B and C authors show that the constructs 1-596 and 1-546 are not interacting with NS5 but it seems that the viral replication is inhibited with 25% in the graph. Please, explain why no statistics were added to these two lanes and address these results.

In figure 5, 6, 7 and 8 authors included DENV instead of ZIKV. This should also be included in the introduction where authors state that they are studying ZIKV and TBEV.

Reviewer #2: Results in Fig S1 are expected in part. LGP2, like RIG-I and MDA5 are ISGs and subsequently upregulated at early times during viral infection. Also, in Fig 1E-F overexpression of LGP2 in CCF-STTG1 cells is likely enhancing MDA5 activity resulting in higher inhibition of viral growth.

L170-1. LGP2 acts as a negative regulator of IFN based on FigS2. This is not clear to me. If both RIG-I and MDA5 are involved in sensing flavivirus infection, as it is believed, the fact that an increase in IFN induction (very mild for some siRNAs) is observed in LGP2 KD cells should be discussed. Is this due to the predominant role of RIG-I versus MDA5 during ZIKV infection? The LGP2 enhancing effect on MDA5 would be depleted also.

Fig S2B does not correlate with the info in the text (L174).

Fig 3C does not clearly support the colocalization of NS3/5 dsRNA and LGP2. It proves that they are all in the cytoplasm. The graphs only show partial overlapping for LGP2 and NS5 but in the right plot LGP2 and NS5 seems to localize more erratically than dsRNA. Also, why NS5 is only in the nucleus in the lowest panel?

Fig 4C. Improve labeling of the figure. Why are two LGP2 IP panels?

Fig 4D. The quality of this figure is low (too much contrast?). Additionally, a control of RNase treatment should be provided.

Fig 6. Polymerase elongation assays are tested over a stretch of 9 nt. Isn´t it too short? The authors used a 17 nt stretch in previous work. Please, discuss this point for those not familiar with these specific assays.

Elongation and pre-elongation assays in Figs 6, 7 and 8. In Fig 6C no effect or very small is detected at LGP2:NS5 ratio 1:1 and 1:3 for DENV while TBEV NS5 seems more sensitive to the presence of LGP2. Apparently no big difference is observed for DENV between mock and LGP2 RD unless a 5:1 ratio is used. I wonder if this proportion is relevant in a physiological context considering that LGP2 is not a very abundant protein in the cell (depending on induction) while viral proteins become extremely abundant over infection. This is a major issue that might be considered and properly discussed.

Statistical significance should be indicated in all the quantification graphs to better estimate the magnitude of differences between the parameters assayed.

In Fig 7C the LGP2 dose effect on DENV NS5 activity is not obvious either.

L381. “Whether ZIKV and TBEV RdRP binding to LGP2 interferes its regulation for RIG-I and MDA5 is still unclear.” It would be interesting indeed to study the effect of binding to NS5 of the different LGP2 constructs and mutants used in the polymerase assays on the IFN response. I guess that could be easily tested in transfection/infection experiments run in parallel in cells that are IFN-competent or not. This would provide information on the relevance of the effect of LGP2 in replication in the infection context.

Reviewer #3: Fig 1E，When verifying the function of the LGP2 protein, an empty vector is not an appropriate control choice since it does not express any protein. It is strongly recommended to use a plasmid expressing GFP as a control instead, as even the expression of some non-specific proteins can to some degree reduce viral replication. Line 148, Fig 1C, although si-3 reduced the expression of LGP2, why viral replication was not enhanced? Fig 1F, Why is there no increasing trend in LGP2 expression (1ug expression plasmid) from 12-36 hours? Furthermore, at 36 hours post-infection, the expression level of Zika protein and RNA is significantly inconsistent in fig 1E and 1F.

Line 239-242, fig 5C，As mentioned, compared with the empty vector, even the lgp2 mutants 1-596 and 1-546 that do not interact with NS5 can significantly inhibit viral replication. Why is there little difference in the inhibition of viral replication between the LGP2 mutant 1-646 that interacts with NS5 and the mutants 1-596 and 1-546 that do not interact with NS5? From these results, it appears that the interaction between LGP2 and NS5 has little impact on viral replication.

**Part III – Minor Issues: Editorial and Data Presentation Modifications**

Reviewer #1: Authors repeatedly say they are measuring IFN signaling when they measure IFNb transcripts please change to IFN induction as the IFN signaling would imply measuring the effect of IFN e.g. ISGs. The authors also say LGP2 is activated at mRNA and protein levels which they do not do they are measuring upregulation of LGP2, please change.

Please add a drawing of NS5 mutants.

In 4B, is this two replicas of the same experiment? Please clarify in the text. How many times were the different experiment repeated in figure 4?

For clarification please add the virus that was used in Figure 4C.

In figure 2D and E it is a bit confusing for the reader why the authors measure IFNβ during SEV (S2B) and not ZIKV or TBEV which are the viruses studied in the paper also, there is nothing mentioned about SEV in the figure legends to S2. It would be good if this could be clarified.

Please change the triangles in fig 3B to be shorter as lane 3 and 6 does not contain NS3-HA and LGP2-Ha respectively.

Reviewer #2: Three different flaviviruses (ZIKV, TBEV or DENV) are used and compared in specific combinations depending on the assay. Please, explain the reason for the corresponding choice in each case.

The fig legends are cryptic. The information provided in general is insufficient. The cell line used and basic data of infection should be mentioned at least.

L86 rewrite sentence for clarity.

L141 “induces” a robust

L185 “traditional”… Conventional?

Reviewer #3: The figure legend lacks a statistical description, such as fig 1, 2 and 5.

Line 54: "include" should be "includes" to agree with "genus"

Line 73: "may strongly impact on virus life cycle" should be "may strongly impact the virus life cycle".

PLOS authors have the option to publish the peer review history of their article (what does this mean?). If published, this will include your full peer review and any attached files.

Reviewer #1: No

Reviewer #2: No

Reviewer #3: No
---

## [Decision Letter · Decision Letter 1]

16 Aug 2023

Dear Dr. Gong,

We are pleased to inform you that your manuscript 'LGP2 directly interacts with flavivirus NS5 RNA-dependent RNA polymerase and downregulates its pre-elongation activities' has been provisionally accepted for publication in PLOS Pathogens.

Best regards,

Alison Kell

Guest Editor

PLOS Pathogens

Sonja Best

Section Editor

PLOS Pathogens

Kasturi Haldar

Editor-in-Chief

PLOS Pathogens

orcid.org/0000-0001-5065-158X

Michael Malim

Editor-in-Chief

PLOS Pathogens

orcid.org/0000-0002-7699-2064

Reviewer Comments (if any, and for reference):

Reviewer's Responses to Questions

**Part I - Summary**

Reviewer #1: The authors have truly answered and done all that the reviewers asked for and this have improved the manuscript substantially.

Reviewer #3: The authors have addressed all my comments and I am happy to recommend publication of the revised manuscript.

**Part II – Major Issues: Key Experiments Required for Acceptance**

Reviewer #1: (No Response)

Reviewer #3: (No Response)

**Part III – Minor Issues: Editorial and Data Presentation Modifications**

Reviewer #1: (No Response)

Reviewer #3: (No Response)

PLOS authors have the option to publish the peer review history of their article (what does this mean?). If published, this will include your full peer review and any attached files.

Reviewer #1: No

Reviewer #3: No

---

## [Editor Report · Acceptance letter]

23 Aug 2023

Dear Dr. Gong,

We are delighted to inform you that your manuscript, " LGP2 directly interacts with flavivirus NS5 RNA-dependent RNA polymerase and downregulates its pre-elongation activities ," has been formally accepted for publication in PLOS Pathogens.

Best regards,

Kasturi Haldar

Editor-in-Chief

PLOS Pathogens

orcid.org/0000-0001-5065-158X

Michael Malim

Editor-in-Chief

PLOS Pathogens

orcid.org/0000-0002-7699-2064